# TRPM8 contributes to sex dimorphism by promoting recovery of normal sensitivity in a mouse model of chronic migraine

David Alarcón-Alarcón[1,2], David Cabañero [1,2] ✉, Jorge de Andrés-López [1], Magdalena Nikolaeva-Koleva [1], Simona Giorgi [1], Gregorio Fernández-Ballester [1], Asia Fernández-Carvajal [1,3] ✉ & Antonio Ferrer-Montiel [1,3] ✉

TRPA1 and TRPM8 are transient receptor potential channels expressed in trigeminal neurons that are related to pathophysiology in migraine models. Here we use a mouse model of nitroglycerine-induced chronic migraine that displays a sexually dimorphic phenotype, characterized by mechanical hypersensitivity that develops in males and females, and is persistent up to day 20 in female mice, but disappears by day 18 in male mice. TRPA1 is required for development of hypersensitivity in males and females, whereas TRPM8 contributes to the faster recovery from hypersensitivity in males. TRPM8-mediated antinociception effects required the presence of endogenous testosterone in males. Administration of exogenous testosterone to females and orchidectomized males led to recovery from hypersensitivity. Calcium imaging and electrophysiological recordings in in vitro systems confirmed testosterone activity on murine and human TRPM8, independent of androgen receptor expression. Our findings suggest a protective function of TRPM8 in shortening the time frame of hypersensitivity in a mouse model of migraine.

Chronic migraine is a highly prevalent and recurrent headache affliction particularly severe in women[1]. While considerable advances have been made in the understanding of the disease[2,3], the mechanisms underlying the sex dimorphism of chronic migraine remain largely unknown. A migraine model with high predictive validity is the sensory sensitization induced by nitroglycerin (NTG)[4]. Acute NTG treatments lead to a delayed hypersensitivity to mechanical stimulation that lasts hours in humans and rodents[4,5]. Furthermore, repeated NTG exposure causes a chronic hypersensitivity that lasts several weeks in murine models[6]. This chronic hypersensitivity is characterized by generalized cutaneous sensitization, which has also been described as a reliable predictor of migraine chronification in humans[7,8] that is found more prevalently in women[9,10].

Multiple migraine triggers including NTG have demonstrated a crucial involvement of transient receptor potential ankyrin 1 (TRPA1) in rodent models of acute migraine[5,11]. In these acute models, NTG acts as a donor of nitric oxide triggering a TRPA1-mediated neuronal activity that promotes nociceptive hypersensitivity[5]. Indeed, TRPA1 is expressed in primary afferent neurons innervating the meninges where its activation favors the release of α-calcitonin gene-related peptide (αCGRP)[11,12], a neuropeptide that plays a pivotal role in migraine development[4,13,14]. Thus, TRPA1 antagonists reduce meningeal inflammation and hypernociception in basic pain models[2,15] and αCGRP inhibition constitutes an effective strategy for migraine treatment in humans[3]. Although recent reports have described higher contribution of αCGRP in females[16], the involvement of TRPA1 in males and females has not been yet studied in preclinical models of chronic migraine.

[1]Instituto de Investigación, Desarrollo e Innovación en Biotecnología Sanitaria de Elche (IDiBE), Universidad Miguel Hernández de Elche, Elche, Spain. [2]These authors contributed equally: David Alarcón-Alarcón, David Cabañero. [3]These authors jointly supervised this work: Asia Fernández-Carvajal, Antonio Ferrer-Montiel. ✉e-mail: dcabanero@umh.es; asia.fernandez@umh.es; aferrer@umh.es

An additional TRP channel tightly associated with the expression of chronic migraine in humans is the transient receptor potential melastatin 8 (TRPM8). Several single-nucleotide polymorphisms affecting TRPM8 have been linked to chronic migraine and allodynia[17,18]. Concurrently, TRPM8 agonists such as menthol have been used medicinally for the alleviation of migraine-related[19,20] and TRPA1-associated pain[21]. TRPM8 is a cation channel expressed in primary afferent neurons, known for being the menthol receptor and the principal detector of environmental cold[22–24]. As such, TRPM8 activity shows modulatory effects on thermal and mechanical hypersensitivity in preclinical models of pain[25]. However, its presence in internal structures kept at euthermic temperature[26] and its recent description in central brain areas[27] suggest additional functions of this protein that may go beyond cold perception. In this line, TRPM8 was previously described as a testosterone receptor in cellular models[28]. While high testosterone levels have been associated with decreased nociceptive sensitivity in mice[29,30] and humans[31], it is unknown whether testosterone-TRPM8 interactions could have functional relevance in nociception.

Here we implemented a murine model of chronic migraine that displays a sexual dimorphism characterized by enhanced nociceptive sensitivity of females as described in humans[3]. The aim of the study was to investigate the involvement of TRPA1 and TRPM8 in this sexually dimorphic behavior. To address this, mechanical sensitivity was assessed in mice of both sexes chronically exposed to NTG and the participation of TRPA1 and TRPM8 was evaluated through genetic and pharmacological approaches. To dissect the functional and molecular consequences of TRPA1 and TRPM8 activities, murine cultures of trigeminal neurons and transfected cell lines expressing murine and human receptors were evaluated through calcium imaging. After finding a male-specific function of TRPM8, the role of testosterone-TRPM8 interactions and the effects of exogenous TRPM8 stimulation were elucidated in in vitro and in vivo models of acute and chronic pain. Molecular docking in human ligand-receptor models may provide further insight on testosterone activity through TRPM8 and the effects of this interaction were corroborated through electrophysiological recordings that show independency of androgen receptor expression. Collectively, our data suggest that testosterone, through its interaction with TRPM8, drives sexual dimorphism in the model of chronic migraine and likely in other pain-related behaviors.

## Results

### Repeated NTG treatment induces a persistent mechanical hypersensitivity that remains up to 20 days in female mice but disappears in male mice

We implemented in our laboratory a modification of a previous model of chronic migraine induced by repeated NTG treatment[6]. NTG (10 mg/kg) or vehicle (5% dextrose and 0.105% propylene glycol in water) were administered intraperitoneally (i.p.) every other day during 8 days to male and female mice, and mechanical sensitivity was assessed before and after each treatment and for 12 additional days after the end of the repeated administrations (Fig. 1a). Two hours after each NTG treatment, acute hypersensitivity was observed in male and female mice, whereas the vehicle did not induce nociceptive sensitization (Fig. 1b, left panel, $p < 0.001$ treatment effect). Mechanical sensitivity assessed before NTG injections and after the end of the repeated treatment (up to 20 days after beginning of the treatments, right panel of Fig. 1b) showed a long-lasting hypersensitivity in male mice (days 2–16, $p < 0.05$ vs. baseline and vehicle, Fig. 1b) that returned to baseline values 8 days after the last NTG injection (day 18, Fig. 1b, nonsignificant vs. baseline or vs. vehicle). In contrast, female mice presented a persistent cutaneous hypernociception that was significant until the end of the experimental procedure ($p < 0.05$ vs. baseline, vs. vehicle, Fig. 1b). Thus, data obtained in this observation period indicate a sexual dimorphism as males can fully recover from NTG sensitization, while females improve partially.

### Repeated NTG treatment induces long-lasting TRPA1-dependent hypersensitivity in female and male mice

Migraine-related nociception produced by acute NTG treatment has been associated to TRPA1 activity in trigeminal ganglia of male mice[5]. We investigated the possible participation of TRPA1 on the model of chronic migraine. Wild-type and TRPA1 knockout mice received NTG injections and mechanical sensitivity was assessed (Fig. 2a). As expected, wild-type mice showed acute hypersensitivity 2 h after each NTG injection. On the contrary, this sensitization was absent in TRPA1 knockout mice of both sexes (Fig. 2a, $p < 0.001$ genotype effect, left panels), in agreement with a previous study describing lack of acute NTG sensitization in TRPA1 knockout males[5]. TRPA1 deletion also prevented the development of chronic hypersensitivity in males and females (Fig. 2a, right panels).

To understand the role of TRPA1 in mediating NTG-induced sensitization in mice, we assessed TRPA1 mRNA expression and intracellular calcium imaging in trigeminal ganglia of male and female mice chronically exposed to NTG (Fig. 2b, c). Samples from NTG-treated wild-type animals showed an increased TRPA1 mRNA expression regardless of the sex (Fig. 2b, $p < 0.001$ vs. vehicle). In parallel, trigeminal neuronal cultures of male and female mice revealed calcium transients in response to the TRPA1 agonist allyl isothiocyanate (AITC) at 70 μM (Fig. 2c). These responses were of similar magnitude in samples of both sexes and had higher amplitude in neurons of mice chronically exposed to NTG ($p < 0.01$ vs. vehicle, Fig. 2d). These results obtained with non-ratiometric calcium imaging were obtained by normalization of AITC-induced transients to 40 mM KCl responses which were similar in vehicle and NTG-treated mice. The neuronal population responding to AITC was composed of small to medium size neurons (100–700 μm², Supplementary Fig. 2a). This increased TRPA1 responsiveness was also characterized by a higher percentage of cells showing significant activity (percentage of KCl-sensitive cells, $p < 0.001$ vs. vehicle, Supplementary Fig. 2b), and the size of these responses was proportional to the percentage of sensitive cells (Supplementary Fig. 2c). Thus, the trigeminal cultures revealed small to medium size neurons with increased TRPA1 activity after chronic NTG treatment, regardless of the sex.

To further investigate TRPA1 involvement on NTG sensitization, trigeminal neurons of naïve wild-type and TRPA1 knockout mice were cultured to assess calcium responses. These neurons were challenged with 100 μM NTG followed by 70 μM AITC (Fig. 2e). A 15.7 ± 3.2% of the cells responded to both stimuli in wild-type mice, while this percentage decreased drastically in TRPA1 knockouts (0.6 ± 0.3%, $p < 0.001$, Fig. 2f), revealing an essential participation of TRPA1 in calcium transients elicited by both NTG and AITC in mice. To assess the translatability of this NTG activity to human TRPA1, NTG was applied to human TRPA1-transfected HEK293 cells and IMR90 fibroblast cells natively expressing this receptor (Fig. 2g). Both cell types showed a dose-dependent relationship for NTG-evoked calcium influx, whereas control HEK293 cells did not respond. Thus, NTG activated both human and murine TRPA1 channels.

TRPA1-mediated trigeminal sensitization may involve the release of the vasodilator peptide αCGRP[11,12]. This neuropeptide is an essential neurotransmitter for migraine neuroinflammation and pain[13]. Hence, we next examined NTG-induced αCGRP release in cultured trigeminal neurons through αCGRP immunofluorescence. A treatment with DD04107, an exocytosis inhibitor that interacts with the exocytosis-related protein SNAP-25[32] was used to investigate vesicular release. Control neurons exposed to NTG vehicle showed stronger αCGRP immunoreactivity than NTG-treated neurons (Fig. 2h, i, $p < 0.001$) suggesting vesicular αCGRP release after NTG. In contrast, neurons pre-treated with DD04107 showed significantly higher αCGRP

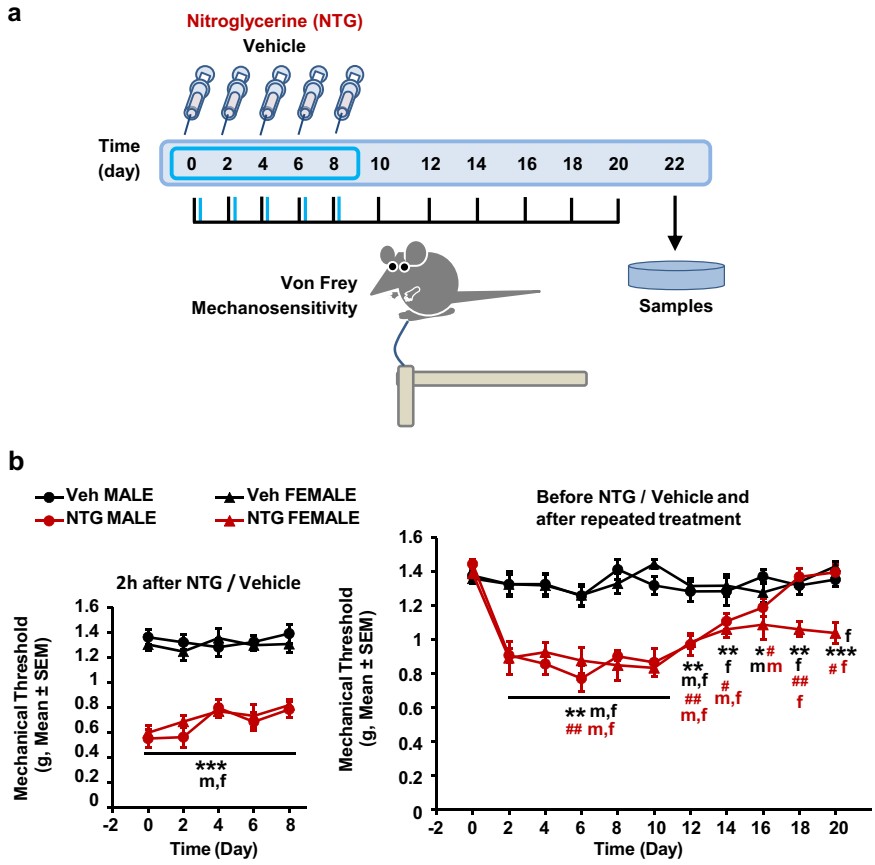

**Fig. 1 | Chronic nitroglycerin administration induces generalized long-lasting mechanical hypersensitivity in male and female mice. a** Model of chronic migraine induced by five intraperitoneal injections of 10 mg/kg nitroglycerin or its vehicle administered every other day (days 0, 2, 4, 6 and 8). Hind paw mechanical sensitivity was assessed with von Frey filaments 2 h after nitroglycerin administration (blue lines), before nitroglycerin injections and after the end of the treatments on days 10, 12, 14, 16, 18 and 20 (black lines). Tissue samples were obtained at the end. **b** Left panels. Nitroglycerin (red) induces similar acute hypersensitivity in males and females, whereas the vehicle (black, 5% dextrose, 0.105% propylene glycol) did not induce nociceptive sensitization. **b** Right panels. Chronic administration of nitroglycerin induces a long-lasting hypersensitivity that is persistent exclusively in females, whereas female or male mice treated with vehicle show normal sensitivity. Data are mean ± S.E.M. **$p < 0.01$, ***$p < 0.001$ vs. vehicle. #$p < 0.05$, ##$p < 0.01$, ###$p < 0.001$ vs. baseline. Two-way RM ANOVA $n = 6$ mice per each vehicle group, $n = 7$ mice per each NTG group. NTG nitroglycerin, Veh Vehicle. Source data provided in the Source Data file, statistical results in Supplementary Table 1.

immunoreactivity (Fig. 2i, $p < 0.01$ vs. NTG + vehicle), indicating inhibition of the neuropeptide release after NTG. Collectively, these data indicate that TRPA1 activity was essential for acute and chronic NTG hypersensitivity, although it could not explain the observed sexual dimorphism.

### TRPM8 activity determines the recovery of normal sensitivity in male mice exposed to the model of chronic migraine

TRPM8 is a thermoTRP channel that has been signaled in the pathophysiology of chronic migraine. Thus, we next investigated the involvement of TRPM8 in NTG chronic sensitization. For this purpose, male and female wild-type and TRPM8 knockout mice were exposed to the chronic NTG treatment (Fig. 3a). Wild-type males presented the expected sensitization (Fig. 3a, $p < 0.01$ vs. Baseline on days 8 and 15) that was resolved by the end of the experimental procedure (day 20, Fig. 3a). Noteworthy, TRPM8 knockout males maintained a persistent sensitization that was significant until the last day of measurements, akin to the persistent sensitization observed in females ($p < 0.01$ vs. Baseline and wild-type, Right panel of Fig. 3a). This finding revealed a protective function of TRPM8 in males subjected to NTG-mediated chronic sensitization and signaled a potential role of this channel in sex dimorphism in the model of chronic migraine.

To investigate the involvement of TRPM8 in sex dimorphism, we next analyzed its expression in trigeminal ganglia of male and female wild-type mice chronically exposed to NTG. TRPM8 mRNA values were similar regardless of treatment and sex (Fig. 3b). Functionally, perfusion with the selective and potent TRPM8 agonist WS12 (500 nM) elicited calcium transients of similar amplitude in both sexes, after chronic vehicle or NTG (Fig. 3c, d). WS12 activated $8 \pm 2.1\%$ of the cultured neurons (Supplementary Fig. 3a), which were small size cells (50–400 $\mu m^2$, Supplementary Fig. 3b, c). This percentage was similar in samples of vehicle and NTG-treated mice. Hence, trigeminal cultures showed that TRPM8 activity and expression were similar between sexes and after the NTG treatment.

Given this similar functionality, we conducted additional experiments to test the antinociceptive efficacy of WS12 in wild-type female mice sensitized after chronic NTG. On day 20, females received WS12 or its vehicle. The observed responses were highly variable and nonsignificant results were found after 10 mg/kg WS12 (i.p.), a dose with reported antinociceptive efficacy in male mice[33] (Fig. 3e). In a separate experiment, we tested the effect of lower (5 mg/kg, Supplementary Fig. 4a) and higher doses of WS12 (20 mg/kg, Supplementary Fig. 4b) in wild-type and TRPM8 knockout females. Both the highest dose (Supplementary Fig. 4b, $p = 0.053$ vs. vehicle), and the lowest dose were ineffective. WS12 showed efficacy when administered in a vehicle containing 5% ethanol and 45% cyclodextrin, however possible antinociceptive effects of this vehicle were detected (Supplementary Fig. 4c). To further characterize the

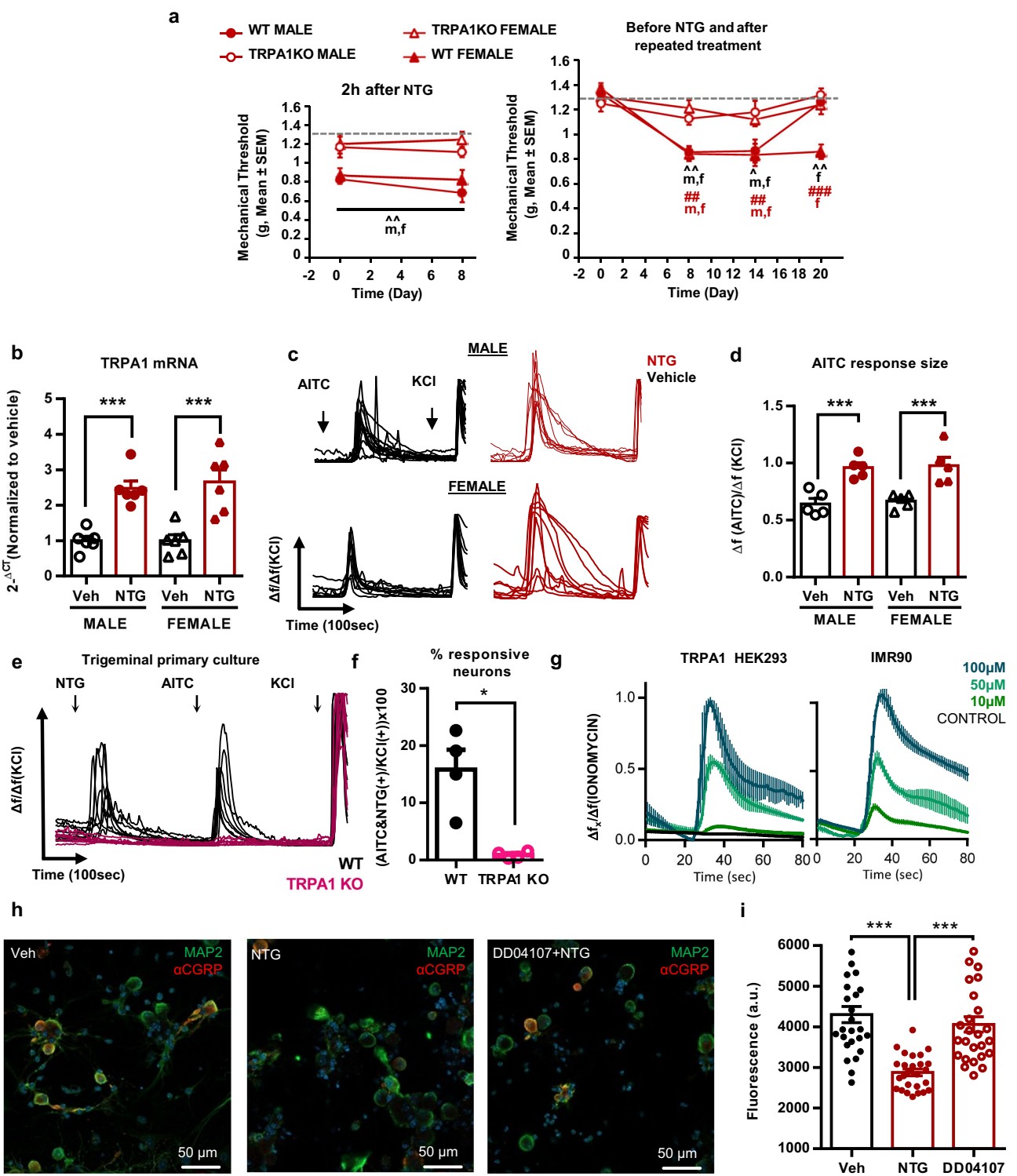

in vivo activity of WS12 in females, we decided to examine the antinociceptive effect of TRPM8 activity on the formalin pain model, another TRPA1-dependent pain model[34]. As illustrated in Fig. 3f, WS12 induced significant antinociception in the acute phase, whereas the late phase was unaltered, revealing a short-lasting effect of this compound.

To further investigate the function of TRPM8 in wild-type males, mice previously exposed to NTG that had recovered their baseline sensitivity were exposed to increasing doses of the TRPM8 blocker AMTB (Fig. 3g). This compound precipitated a significant re-sensitization when administered at 10 or 15 mg/kg

($p < 0.05$ vs. vehicle, Fig. 3g) in this model. Interestingly, similar results were obtained in male mice previously subjected to the formalin test. To corroborate the existence of a TRPM8-dependent tone in males providing relief after TRPA1-induced nociception, male mice were treated with AMTB after the extinction of formalin-induced nocifensive activity. These mice showed a significant reinstatement of licking behavior evident 30 min after administration of the compound ($p < 0.05$ vs. vehicle, Fig. 3h). Altogether, these results suggest the presence of endogenous TRPM8 activity with antinociceptive function in male mice.

**Fig. 2 | TRPA1-related activity is essential for mechanical hypersensitivity after nitroglycerin and TRPA1 expression and function increase in trigeminal ganglia of chronically exposed males and females. a** TRPA1 deletion prevents acute and chronic mechanical hypersensitivity after nitroglycerin in male and female mice. **b** Trigeminal ganglia of male and female mice show increased TRPA1 mRNA expression after chronic nitroglycerin exposure. **c** Representative calcium traces in response to 70 μM AITC normalized to 40 mM KCl in trigeminal cells of mice chronically exposed to vehicle or nitroglycerin. **d** AITC responses increase in trigeminal cultures from nitroglycerin-treated animals regardless of the sex. Average response size of AITC-sensitive neurons vs. KCl responses. **e** Wild-type trigeminal neurons responding to 100 μM nitroglycerin also respond to 70 μM AITC. TRPA1 knockout mice lack these responses. **f** 15% of trigeminal neurons respond to AITC and NTG in wild-type mice vs. 1–2% in TRPA1 knockouts. **g** TRPA1-transfected HEK293 cells and IMR90 cells constitutively expressing hTRPA1 show calcium responses after 10, 50 and 100 μM nitroglycerin. **h** Trigeminal culture exposed to vehicle, nitroglycerin or the exocytosis inhibitor DD04107. Cells labeled with pan-

neuronal marker antiMAP2 (green), antiαCGRP (red) and nuclei marked with DAPI (blue). **i** Nitroglycerin decreases intracellular αCGRP immunoreactivity and DD04107 prevents this. Mean mechanical thresholds (**a**), expression (**b**), calcium response (**d**, **f**) or red fluorescence (**i**) ± S.E.M. Datapoints without error bars represent values of individual animals, except for **i** where dots represent quantified images. **a** ^$p < 0.05$, ^^$p < 0.01$, ^^^$p < 0.001$ vs. vehicle. #$p < 0.05$, ##$p < 0.01$ vs. baseline, two-way RM ANOVA, $n = 6$ mice per condition except for WT female group which was of five mice; **b** ***$p < 0.001$ two-way ANOVA, $n = 6$ samples from six different mice per condition, obtained after one behavioral experiment; **d** ***$p < 0.001$ two-way ANOVA, $n = 5$ samples from five different mice per condition, obtained in three independent experiments; **f** *$p < 0.05$ two-sided Mann–Whitney $U$ $n = 5$ samples from five different mice per condition, obtained in three independent experiments; **i** ***$p < 0.001$, one-way ANOVA $n = 26$ images from samples of five mice per condition, obtained in five independent experiments. NTG nitroglycerin, Veh vehicle. Source data are provided in Source Data file, statistical results in Supplementary Table 2.

## Testosterone activates TRPM8 to resolve mechanical sensitization in mice exposed to the model of chronic migraine

Testosterone has been suggested as an endogenous TRPM8 agonist[28] and shows antinociceptive functions in mice[29]. We hypothesized that this endogenous androgen could have a protective function in the mouse model of chronic migraine. To address this question, mice were first subjected to a sham surgery or to an orchidectomy to deplete gonadal testosterone. Once their nociceptive sensitivity was restored the animals received the NTG treatment (Fig. 4a). Acute NTG produced similar hypernociception in sham and orchidectomized animals (Fig. 4a, left panel, $p < 0.001$ vs. vehicle). In contrast, the repeated NTG evidenced a persistent chronic sensitization selectively expressed in orchidectomized animals (Fig. 4a, right panel, $p < 0.01$ vs. sham), indicating possible protective role of testosterone.

To investigate whether testosterone could have TRPM8-mediated antinociceptive effect in males, wild-type and TRPM8 knockout males were orchidectomized and received subcutaneous osmotic pumps filled with testosterone or its vehicle (cyclodextrin 45% in water). Afterwards, all mice were chronically treated with NTG, and their mechanical sensitivity was assessed. Testosterone induced complete recovery of mechanical thresholds in wild-type mice (Fig. 4b, right panel, day 20, $p < 0.001$ vs. Vehicle wild-type, nonsignificant vs. baseline), while vehicle-treated animals remained sensitized by the end of the experiment (Fig. 4b, right pane, $p < 0.001$ vs. baseline on day 20). In contrast, TRPM8 knockout mice lacked this restorative effect of testosterone (Fig. 4b, right panel, day 20, $p < 0.001$ vs. wild-type, $p < 0.001$ vs. baseline), although an antinociceptive effect independent of TRPM8 activity was also evidenced in knockouts (Fig. 4b, right panel, day 20, $p < 0.05$ vs. vehicle TRPM8 knockout). To further clarify the involvement TRPM8 in the protective effect of testosterone, AMTB was administered to all mice after the nociceptive measurement on day 20 (Fig. 4c). A significant drop in the mechanical thresholds was selectively observed in wild-type animals treated with testosterone (Fig. 4c, $p < 0.01$ vs. values before AMTB), whereas mice receiving vehicle and knockouts showed unaltered mechanosensitivity. These results revealed significant testosterone-TRPM8 antinociception in males.

Next, to determine whether testosterone could have rapid antinociceptive effects we treated NTG-exposed wild-type and TRPM8 knockout females with a single subcutaneous administration of 1 mg/kg testosterone or its vehicle (2.5% DMSO in corn oil, Fig. 4d). A significant testosterone-induced antinociception was observed selectively in wild-type mice ($p < 0.05$ vs. pre-treatment values, $p < 0.05$ vs. testosterone-treated TRPM8 knockouts). Overall, the present behavioral results suggest nongenomic antinociceptive effects of testosterone through TRPM8.

We next investigated if testosterone activated TRPM8 channels in trigeminal cultures as a mechanism to account for the protective role

of the androgen. Trigeminal cultures of wild-type mice showed calcium transients in response to 10 pM testosterone (Fig. 4e). Cells responding to testosterone also presented calcium transients in response to 500 nM WS12 (7.9 ± 0.8% of KCl-sensitive cells; Fig. 4e). These responses were abolished in neural cultures from TRPM8 knockout animals tested side-by-side (Fig. 4e, $p < 0.001$ vs. wild-type), revealing a testosterone activity through TRPM8 in trigeminal neurons. Previous studies described structural and functional differences between murine and human TRPM8[35]. To evaluate this possibility, we assessed testosterone activity on HEK293 cells constitutively expressing murine or human TRPM8 (Fig. 4f). Similar calcium transients were elicited after testosterone 10 pM in cells expressing human or murine TRPM8, whereas control HEK293 cells lacked this response (Fig. 4f). Altogether, the present data reveal a testosterone activity on murine and human TRPM8.

## Testosterone-induced current of human TRPM8 is independent of androgen receptor expression

Given the possible interaction of testosterone with TRPM8, we conducted patch-clamp whole-cell recordings to assess possible testosterone-induced currents on HEK293 cells heterologously expressing human TRPM8 (hTRPM8-HEK293). Current density (pA/pF) vs. voltage relationships in these cells (Fig. 5a) evidenced an increase in outward rectifying current after 10 s application of 10 pM testosterone. This testosterone-induced current was significantly different from the basal current (Fig. 5b, $p < 0.05$) and was completely blocked in the presence of 10 μM AMTB, the specific TRPM8 blocker (Fig. 5b, $p < 0.01$ vs. 10 pM testosterone). As testosterone's primary mechanism of action is through binding to the androgen receptor (AR), an additional experiment was conducted knocking down the androgen receptor in hTRPM8-HEK293 cells. Thus, hTRPM8-HEK293 cells were transfected with small interfering RNA (siRNA) targeting the AR or with scrambled siRNA. In parallel, wild-type HEK293 cells lacking heterologous expression of TRPM8 were assayed (WT HEK293). hTRPM8-HEK293 cells transfected with either anti-AR siRNA or scrambled siRNA showed similar current density-voltage curves after 10 pM testosterone (Fig. 5c), whereas WT HEK293 cells did not display testosterone-induced currents ($p < 0.01$ WT HEK293 vs. hTRPM8-HEK293, current fold increase at +80 mV, Fig. 5d). To ensure the silencing of AR expression, western blot was performed in samples of hTRPM8-HEK293 and WT HEK293 cells cultured in parallel to the cells used for electrophysiological recordings (Fig. 5e). A disruption of AR expression was confirmed in hTRPM8-HEK293 cells treated with anti-AR siRNA, whereas hTRPM8-HEK293 cells treated with scrambled siRNA or WT HEK293 cells presented evident AR expression. In order to allow the comparison to an established TRPM8 agonist in the same setting, we conducted dose-response curves for testosterone and WS12 in untreated hTRPM8-HEK293 cells, obtaining EC50s of 7 pM for

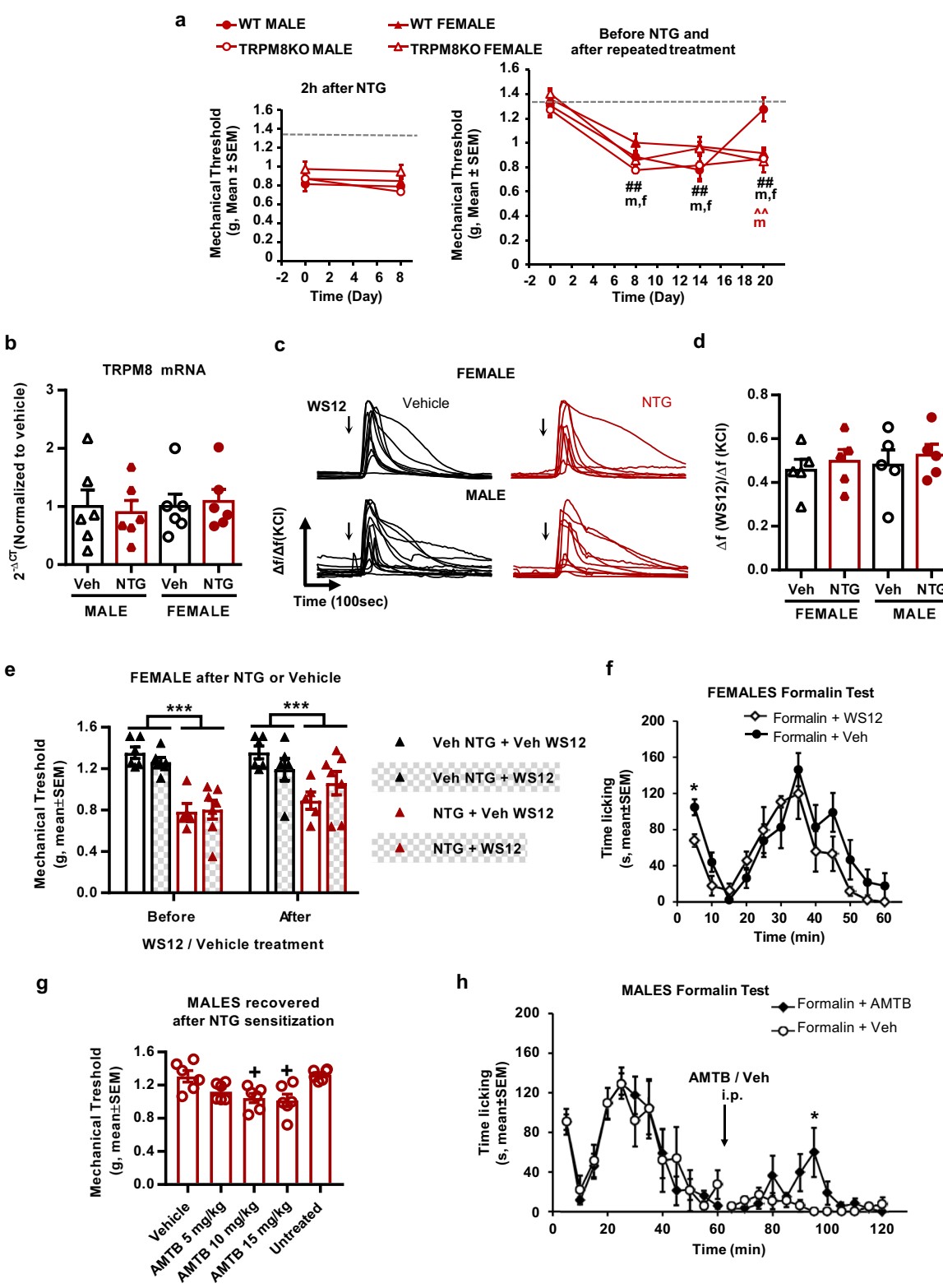

testosterone and 500 nM for WS12 (Supplementary Fig. 5a). The EC50 of WS12 was afterwards used to estimate the IC50 of the TRPM8 antagonist AMTB, which also binds to the menthol-binding site[36] (Supplementary Fig. 5b). Subsequent application of AMTB IC50 (15 nM) shifted the testosterone dose-response curve to the right (Fig. 5f, EC50 = 34 pM), being compatible with competitive inhibition. Hence, testosterone-induced currents were selectively elicited in

HEK293 cells heterologously expressing human TRPM8, were independent of AR expression and sensitive to the TRPM8 antagonist AMTB, substantiating an ionotropic action of the hormone[28].

## Discussion

The most salient contribution of our study is the discovery of the role of TRPM8 as an androgen-activated receptor that provides

**Fig. 3 | TRPM8 allows recovery of normal sensitivity in male mice exposed to nitroglycerin or formalin. a** Wild-type and TRPM8 knockout males display similar acute mechanical hypersensitivity after nitroglycerin (left panel). Deletion of this receptor prevents recovery of baseline mechanical sensitivity after chronic nitroglycerin treatment in males (right). Acute and long-lasting hypersensitivity is similar in wild-type and knockout females. **b** Trigeminal TRPM8 mRNA expression is similar regardless of sex or nitroglycerin/vehicle treatment. **c** Calcium responses of trigeminal cultures from males and females chronically exposed to nitroglycerin or vehicle after exposure to TRPM8 agonist WS12 (500 nM). **d** Average size of calcium transients after WS12 is similar in neurons from males and females. Response sizes vs. respective KCl responses. **e** Female mice chronically exposed to nitroglycerin show increased mechanical sensitivity when compared to vehicle-treated mice. This sensitivity remained unchanged after 10 mg/kg WS12 i.p. **f** Nocifensive behavior in the acute phase of the formalin test (5 min) is significantly alleviated in females receiving 6 nmol WS12 i.pl. **g** TRPM8 antagonist AMTB (i.p.) unmasks latent mechanical pain sensitization in mice chronically exposed to nitroglycerin that already recovered their basal sensitivity. **h** 10 mg/kg AMTB administered 1 h after formalin injections induces significant reinstatement of licking behavior in males. Mean mechanical thresholds (**a, e, g**), expression (**b**), response size (**d**), nocifensive behavior (**f, h**) ± S.E.M. Datapoints without error bars represent values of individual animals. **a** ##$p < 0.01$ vs. baseline. ^^$p < 0.01$ vs. vehicle two-way RM ANOVA $n = 5$ mice per condition. **b** Two-way ANOVA $n = 6$ samples from six different mice per condition, obtained after one behavioral experiment. **d** Two-way ANOVA $n = 5$ samples from five different mice per condition, obtained after three independent experiments. **e** ***$p < 0.001$ nitroglycerin vs. vehicle, three-way ANOVA $n = 6$(Veh + Veh), $n = 5$(NTG + Veh), $n = 6$(Veh + WS12), $n = 7$(NTG + Veh WS12) mice. **f** *$p < 0.05$ vs. vehicle. Multiple two-sided $t$-test $n = 4$(WS12) or 7(Vehicle) mice. **g** +$p < 0.05$ vs. vehicle. Two-sided Friedman test $n = 6$ mice. **h** *$p < 0.05$ vs. vehicle. Multiple two-sided $t$-test $n = 6$ mice per condition. ORX orchidectomy, NTG nitroglycerin, Veh vehicle. Source data are provided in Source Data file, statistical results in Supplementary Table 3.

antinociception in a sex-dependent fashion. Specifically, TRPM8 activity favors recovery in a mouse model of NTG-induced chronic migraine that produces similar acute mechanosensitivity in males and females but persistent chronic hypersensitivity exclusively in females. Our results suggest that testosterone by activating TRPM8 channels drives a sexual dimorphism characterized by recovery of normal sensitivity in males. We propose that the lack of this protective mechanism in females leads to a persistent mechanical hypersensitivity. Noteworthy, our model of chronic migraine mimics the sexual dimorphism observed in humans, characterized by stronger transitions to chronic sensitization in women[30] and possible faster recovery in men. Previous studies also described higher female sensitivity in models of formalin-inflammatory pain[37] and models of persistent pain such as stress-induced visceral hypersensitivity or muscle pain after repeated saline injections and forced activity[29,37,38]. We considered the direct assessment of trigeminal sensitization in the head, however the repeated restraint stress required for this type of assessment could have been an additional factor in promoting sexual dimorphism[39]. Overall, these findings suggest that the exposure to repeated noxious insults such as ongoing inflammation and stress-related stimuli favor the perpetuation of painful responses in females, whereas males show faster recovery and are able of reinstating their normal sensitivity.

In our model of chronic migraine, NTG provokes mechanical hypersensitivity by signaling through the TRPA1 channel, although this receptor is not involved in the sex dimorphism observed. Indeed, repeated NTG exposure induced similar TRPA1 mRNA overexpression in trigeminal ganglia of male and female mice. Increased expression translated into exacerbated TRPA1 activity and trigeminal cultures of males and females chronically exposed to NTG presented stronger TRPA1 activity in response to the specific agonist AITC. In line with our results, previous research described TRPA1-specific neuronal responses after acute NTG[5], and male rodents exposed to NTG were sensitive to TRPA1 antagonism[40]. Our findings reveal an essential participation of TRPA1 for the development of chronic NTG hypersensitivity in male and female mice, in agreement with studies showing TRPA1 involvement in persistent nociceptive sensitizations[12,41,42].

The NTG-TRPA1 mechanism involved in promoting the mechanical hypersensitivity relies on the release for αCGRP from trigeminal neurons[11,12], in agreement with the critical role of this neuropeptide in the aetiology of chronic migraine[13]. Our data further substantiate this tenet as sensitivity to the exocytosis inhibitor DD04107 revealed a mechanism involving large dense core vesicles and vesicular fusion protein SNAP-25[32]. Notably, the data obtained with DD04107 suggest a potential efficacy of this peptide for the treatment of chronic migraine through inhibition of αCGRP release, similar to botulinum neurotoxin[14]. Accordingly, we propose antagonistic roles of TRPA1 and TRPM8 activities determined by the continuous exposure to exogenous and endogenous agonists. Namely, nitric oxide derived from the treatment with NTG elicits a hyperalgesic state via TRPA1 stimulation both in males and females, whereas testosterone exerts an antinociceptive role through interactions with TRPM8. Several mechanisms could underlie TRPM8-mediated antinociception. On the one hand, it has been proposed that TRPM8 activation produces silencing of C nociceptors expressing TRPA1[43] through interneuron recruitment in the spinal cord dorsal horn[44,45] or via inhibitory metabotropic glutamatergic receptors expressed in the central terminal of nociceptors[46]. Thus, in the studied pain models TRPM8 activity may occlude signaling of TRPA1-expressing nociceptors and subsequent CGRP release[43]. In addition, a participation of central brain areas cannot be excluded from our data, given the recent description of TRPM8 also in supraspinal locations[27].

Notably, our results in TRPM8 knockout mice subjected to chronic NTG sensitization reveal a protective function of TRPM8. In line with our data, TRPM8 stimulation showed efficacy alleviating thermal hyperalgesia and nocifensive behaviors in models of headache-related pain[44,45]. A protective function was similarly described in models of noxious heat or chemically-induced nociception such as the injection of capsaicin or the TRPA1 agonist acrolein[24,43]. TRPM8 stimulation also provided alleviation of mechanical and cold sensitivity in models of chronic neuropathic pain[47,48]. In accordance with these preclinical findings, TRPM8 agonists have been used medicinally for alleviating a variety of pain conditions including migraine[19,20,49], although using these compounds is not a first-line treatment for this clinical condition[14]. Our results may be in contrast with experiments recently published after the preprint of the present work in which TRPM8 is depicted as a trigger of nociceptive sensitization in the model of chronic migraine[50]. While this work also stresses the relevance of TRPM8 for future investigations, we find experimental differences also involving NTG-induced sensitization that may explain the diverging results. Firstly, their exacerbated hypernociception may be associated with the combined use of NTG with propylene glycol as suggested by our data (Fig. 1 vs. Supplementary Fig. 1). Secondly, the NTG-induced hyperalgesia elicited in our model would have been overlooked in animals with a basal hypersensitive phenotype when compared to the mice used in our experimental conditions[50]. Possible explanations to basal hyperreactivity include substrain differences or lack of habituation. Contradictory results were also published showing pro-algesic effects of TRPM8 agonists on migraine, although the use of TRPM8 agonists with partial effect on TRPA1 such as icilin could have yielded misleading results[51]. Similarly, we did not observe significant antinociception after administration of high WS12 doses in female mice treated with NTG. The absence of significant antinociception after WS12 could be associated to the potent and short-lasting effect of this compound, as suggested by the results obtained in the formalin test where significant effects lasted only 5 min in the acute phase. In agreement, calcium-dependent desensitization of TRPM8 is described

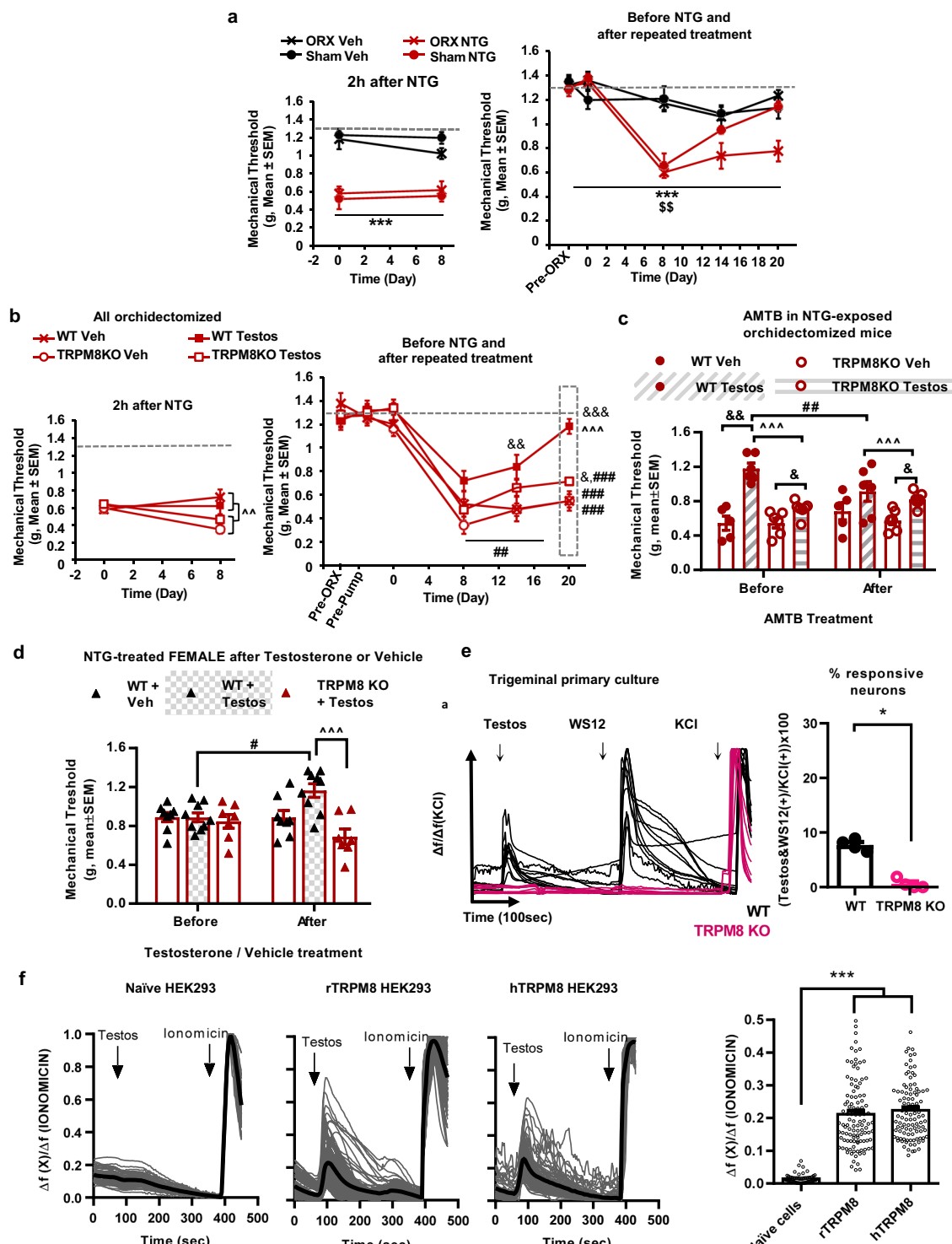

after application of the canonical TRPM8 agonists menthol, icilin or WS12[52]. In our von Frey experiments, reliable assessment of mechanosensitivity requires spaced and sequential application of von Frey filaments during periods of 15–30 min. Hence, mechanosensitivity could not be precisely assessed at that specific temporal resolution and a transient effect of WS12 could have been overlooked. In this line, short-lasting pain-relieving responses to menthol and its derivatives can be found clinically and were described elsewhere[49]. On the other hand, the poor hydrosolubility of WS12 required its dissolution in corn oil for systemic administration and the lack of a significant effect could also be related to an erratic absorption and distribution of the compound.

The protective role of TRPM8 was ratified by the AMTB-induced resensitization of males already recovered from the chronic NTG treatment. This protective TRPM8 activity was corroborated in males previously subjected to the classical formalin test and is compatible with a latent nociceptive sensitization masked by tonic TRPM8 activity. The development of latent nociceptive sensitizations or hyperalgesic priming has been related to the establishment of chronic pain conditions[53–55] and an endogenous opioid tone has been associated with this tonic inhibition of nociception[54]. In this context, an interdependency between TRPM8 and the endogenous opioid system has been described, with TRPM8 knockout mice displaying reduced morphine analgesia and the antinociceptive effects of TRPM8 agonists

**Fig. 4 | Testosterone stimulates neuronal TRPM8 to promote recovery of normal mechanosensitivity after chronic nitroglycerin treatment.**
**a** Orchidectomized and sham mice display similar acute hypersensitivity after nitroglycerin (left panel), however only orchiectomized mice remain sensitized on day 21 (right). **b** Left, all orchidectomized mice show similar acute hypersensitivity after a first nitroglycerin dose, whereas after the last dose, orchidectomized TRPM8KO mice develop stronger sensitization. Right, mice receiving testosterone supplementation show inhibition of the long-lasting hypersensitivity induced after chronic nitroglycerin. This attenuation allows recovery of baseline sensitivity in wild-type mice, but not in TRPM8KO mice. **c** Testosterone-supplemented wild-type mice that recovered normosensitivity reinstate sensitization after AMTB on day 21. This response is absent in TRPM8 knockouts **d** Sensitized wild-type females recover basal sensitivity after acute testosterone treatment on day 21. **e** Cultured trigeminal neurons showing calcium transients after 10 pM testosterone also respond to 500 nM WS12 whereas TRPM8 knockouts are unresponsive. Right, 7% of neurons from wild-type trigeminal cultures show calcium transients after 10 pM testosterone and 500 nM WS12. This is abolished in TRPM8 knockouts. **f** 10 pM testosterone

elicits calcium transients in HEK293 cells when heterologously expressing rat or human TRPM8. Mean mechanical thresholds (**a**–**d**), response size (**e**, **f**) ± S.E.M. Datapoints without error bars represent values of individual animals. **a** ***$p < 0.001$ vs. vehicle, $^{\$\$}p < 0.01$ vs. sham, three-way RM ANOVA. $n = 6$ (Sham-Vehicle, Sham-NTG), $n = 5$ (ORX-Vehicle), $n = 7$ (ORX-NTG) mice. **b** $^{\wedge\wedge}p < 0.01$, $^{\wedge\wedge\wedge}p < 0.001$ vs. TRPM8KO; $^{\&}p < 0.05$, $^{\&\&}p < 0.01$ vs. testosterone vehicle; $^{\#\#}p < 0.01$, $^{\#\#\#}p < 0.001$ vs. baseline, three-way RM ANOVA. $n = 5$ (WT-Veh), $n = 6$ (WT-Testos and TRPM8KO-Veh), $n = 7$ (TRPM8KO-Testos) mice. **c** $^{\wedge\wedge\wedge}p < 0.001$ vs. TRPM8KO; $^{\&}p < 0.05$, vs. testosterone vehicle; $^{\#\#}p < 0.01$ vs. basal, three-way RM ANOVA $n = 5$ (WT Veh), $n = 6$ (WT Testos and TRPM8KO Veh), $n = 7$ (TRPM8KO Testos) mice. **d** $^{\wedge\wedge\wedge}p < 0.001$ vs. TRPM8KO, $^{\#}p < 0.05$ vs. basal, two-way ANOVA $n = 7$–9 mice per condition. **e** *$p < 0.05$ two-sided Mann–Whitney $U$ $n = 4$ samples from four different mice per condition obtained in two independent experiments. **f** ***$p < 0.001$ vs. control, two-sided Mann–Whitney $U$ $n = 64$–100 cells of two independent cultures per group. ORX orchidectomy, NTG nitroglycerin, Veh vehicle, Testos testosterone, TRPM8KO TRPM8 knockout. Source data are provided in Source Data file, statistical results in Supplementary Table 4.

being sensitive to the opioid antagonist naloxone[33,56]. Our results suggest the interest of investigating opioid-TRPM8 interactions for the promotion of antinociception.

Noteworthy, testosterone application provided rapid TRPM8-mediated antinociception in females. In line with this finding, orchidectomized males showed persistent sensitization after chronic NTG similar to females. Previous studies also described decreased ability of orchidectomized males in restoring normal mechanosensitivity after inflammatory or stress-related insults[29,37,38]. These studies elucidated testosterone antinociceptive mechanisms including transformation to di-hydrotestosterone and binding to androgen receptor[37], down-regulation of anti-opioid neurotransmitter Brain-derived Neuro-trophic Factor (BDNF)[38] or modulation of serotonin transporters[29]. In our study, orchidectomized wild-type males exposed to testosterone recovered their normal sensitivity after cessation of the NTG treatment, and this effect was largely dependent on TRPM8. These data reveal a function of TRPM8 providing endogenous antinociception in males though testosterone stimulation. Male-specific alterations have been previously observed in TRPM8 knockouts, including delayed cold acclimation and lower bone mineral density, similar to females of either genotype that showed this same phenotype[57]. Interestingly, these features are also tightly linked to the activity of sexual steroids[58]. We observed rapid antinociceptive effects of testosterone in NTG-exposed females in a time frame in which acute testosterone anxiolytic effects are also found[59]. In agreement, preclinical studies show anti-nociceptive efficacy of testosterone in male and female rodent models of acute and chronic pain[29,37,38]. Consistently, clinical treatments with testosterone may provide pain alleviation[60], and testosterone levels of men and women are inversely proportional to pain perception[61] including migraine pain[31]. However, hormonal treatments are sub-jected to tight regulation and have transcriptional effects and dele-terious consequences that limit their use.

Our trigeminal cultures revealed TRPM8 calcium transients in response to picomolar concentrations of testosterone[62] that were absent in cultures of TRPM8 knockouts. Whole cell patch-clamp recordings in HEK cells heterologously expressing human TRPM8 also revealed ionic currents at low picomolar testosterone concentrations (EC50 = 7 pM), in contrast to the lower potency of WS12 (EC50 = 500 nM) obtained in the same experimental conditions. These testosterone-induced currents were independent of androgen receptor expression and sensitive to the TRPM8 antagonist AMTB, as revealed by the 5-fold decrease in testosterone potency (EC50 increased from 7 to 34 pM) in the presence of 15 nM AMTB (IC50 of AMTB against the less potent WS12). Note that the high potency of testosterone evoking TRPM8 ionic currents, requires higher AMTB concentrations to abolish testosterone-induced TRPM8 responses. Indeed, 10 μM AMTB fully blocked testosterone-evoked ionic currents.

In agreement with our data, other groups have revealed TRPM8-mediated currents after application of picomolar concentrations of testosterone to cancer cells, somatosensory neurons[28] or planar lipid bilayers expressing TRPM8[63]. Thus, testosterone induced rapid non-genomic effects on primary afferent neurons and other cell types through TRPM8. Similarly, other endogenous molecules such as oxy-tocin have shown agonistic effects over TRPV1 also at picomolar concentrations[64]. While WS12 in the nanomolar range elicits stronger calcium transients than picomolar concentrations of testosterone, at the picomolar level TRPM8 agonists such as menthol, icilin or WS12 lack significant effects on intracellular calcium imaging[28]. On the other hand, nanomolar concentrations of testosterone inhibited TRPM8[65] in HEK cells and neurons, suggesting possible inverted u-shaped dose-effect curves for this type of steroids. In agreement, picomolar con-centrations of dihydrotestosterone induced pronounced over-expression of TRPM8 in cancer cells whereas higher concentrations had a reduced effect[66]. Our calcium imaging, electrophysiological and behavioral results suggest an ionotropic effect of testosterone on TRPM8 channels and hint to the presence of a potential hormone binding site within the receptor structure. Docking simulations on a computational homology model of the human TRPM8 suggest the presence of a putative testosterone binding site in the receptor transmembrane domain with a theoretical binding energy and affinity consistent with the hormone potency (Supplementary Fig. 6a, b and Computational_files.zip). Furthermore, this putative ligand-receptor complex model suggests binding of testosterone to the active pocket described for WS12 and menthol[52,67]. Interestingly, female hormones such as estradiol or progesterone were also able to be positioned in this binding pocket (Supplementary Fig. 6c, d and Computational_fi-les.zip) although with lower binding energy estimates than testoster-one. This agrees with the potency of estradiol and progesterone evoking TRPM8 ionic currents in planar lipid bilayers[63], as compared with testosterone. This computational docking model provides a tes-table hypothesis for future studies aimed to identify the molecular determinants of this putative binding site, either by structural and/or by structure-function studies. In this line TRPM3, another TRP channel of the melastatin family, has been established as a thermoreceptor[68] that also responds to sexual hormones such as pregnenolone and progesterone[69], suggesting a dual role of these protein types as detectors of both exogenous and endogenous stimuli. Our data indi-cate a testosterone-TRPM8 interaction that could be helpful for the design of novel compounds mimicking this channel agonist activity without the unwanted effects of hormonal treatments.

In conclusion, a high sensitization level of females has been found in a model that reproduces the sexual dimorphism and the enhanced cutaneous sensitivity associated with chronic migraine in humans[7] and especially in women[9,10]. This type of chronic sensitization could be

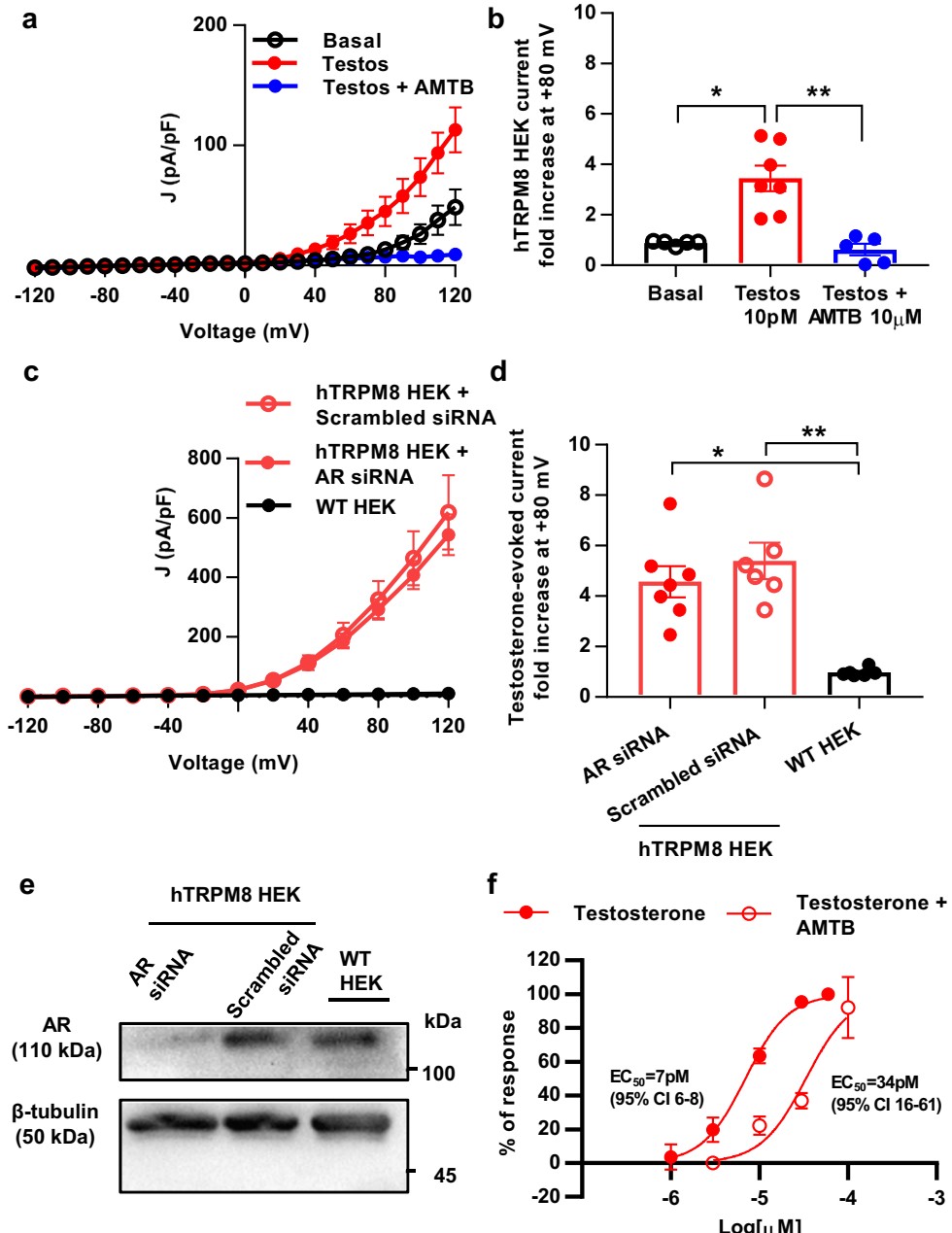

**Fig. 5 | Testosterone increases outward current of heterologously expressed human TRPM8, independently of androgen receptor expression. a** Current density (pA/pF) vs. voltage relationship evidences increase in outward rectifying current 10 pM testosterone in HEK293 cells expressing human TRPM8 (hTRPM8). 10 μM AMTB inhibits testosterone effect ($n = 5$ cells from two independent cultures. Each cell received a testosterone pulse followed by a pulse of AMTB with testosterone). **b** 10 pM testosterone increases basal TRPM8 current at +80 mV ($n = 7$ cells from two independent cultures; $p < 0.017$ vs. Basal current) and 10 μM AMTB reverses its effect ($n = 5$ cells from two cultures, $p < 0.009$ vs. 10 pM testosterone). **c** hTRPM8 HEK 293 cells (hTRPM8 HEK) transfected with small interfering RNA (siRNA) against androgen receptor (AR) ($n = 7$ from two cultures) and control hTRPM8 HEK transfected with scrambled siRNA ($n = 6$ from two cultures) show similar testosterone-induced outward rectifying current. This current is absent in wild-type HEK293 cells (WT HEK, $n = 6$ from 1 culture) lacking TRPM8. **d** At +80 mV, testosterone-evoked current is similar in anti-AR siRNA or scrambled siRNA-transfected hTRPM8 HEK cells, and this current is absent in wild-type HEK

($p < 0.046$ vs. anti-AR siRNA, $p < 0.005$ vs. scrambled siRNA). **e** Western blot of AR (110 kDa) and β-tubulin loading control (50 kDa) shows disrupted AR expression in anti-AR siRNA-transfected hTRPM8 HEK cells. WT HEK and scrambled siRNA-transfected hTRPM8 HEK cells display similar AR expression. **f** 15 nM AMTB (IC50 for WS12 inhibition, Supplementary Fig. 5) shifts to the right the dose-response curve for testosterone-induced currents. EC50 for testosterone (7pM, 95% CI 6–8) increases in presence of AMTB (34 pM, 95% CI 16–61, $n = 9$ cells from three independent cultures for both dose-response curves). "$n$" indicates number of registered cells. **b, d** Data were subjected to Shapiro–Wilk normality tests, then Kruskal–Wallis followed by Mann–Whitney $U$ tests were applied. **$**p < 0.01$, *$p < 0.05$. Data expressed as mean ± SEM, dots on bar charts represent individual cells. EC50s were estimated fitting data to a sigmoidal dose-response curve with constraints (Top = 100, Bottom = 0, GraphPad Prism Software). **e** is a representative western blot of two independent transfections. Source data are provided in Source Data file, statistical results in Supplementary Table 5.

aggravated after recurrent exposure to environmental factors involving TRPA1 stimulation such as pungent substances, oxidative stress or cyclic proinflammatory events[70]. The present data reveals that the difference between males and females is mainly due to an increased antinociception of males. This effect relies partly on TRPM8 activity in response to endogenous testosterone, an androgenic hormone that is present at much higher levels in male individuals. Hence, novel molecules mimicking testosterone activity on TRPM8 could lack the unwanted effects of hormonal treatments and may provide effective pain relief for individuals with low testosterone levels of any gender. In addition, our data endorse the interest of analgesic drugs inhibiting TRPA1 activity for the treatment or prevention of migraine and pain syndromes characterized by enhanced mechanosensitivity in males and females[2,43,71], although careful modulation may be desirable since TRPA1 shows neuroprotective effects as a hypoxia sensor that drives vasodilation and reduces ischemic damage[72].

## Methods

### Animals
Adult male and female mice with a C57BL/6J background (Envigo, Horst, The Netherlands), wild-type or defective in *Trpa1*[73] or *Trpm8*[23] were bred in the animal facility at Universidad Miguel Hernández (UMH, Elche, Alicante, Spain). Experiments involving solely wild-type C57BL/6J mice or TPRA1 knockout mice and their wild-type controls included 8–15-week-old age-matched mice, whereas experiments involving TRPM8 knockouts and their respective wild-type controls included age-matched groups of 8–20 week-old mice. TRPM8 knockout mice were a gift from Dr. F. Viana (Instituto de Neurociencias de Alicante, Alicante, Spain). Care was taken to minimize the number of animals used and the stress they experienced. All experimental procedures were approved by the Animal Care and Use Committees of Universidad Miguel Hernández and the regional government (code: 2018/VSC/PEA/0250-3) and were conducted according to the ethical principles of the International Association for the Study of Pain for the evaluation of pain in conscious animals[74], the European Parliament and the Council Directive (2010/63/EU) and the Spanish law (RD 53/2013). Housing conditions were maintained at $21 \pm 1\,°C$ and $55 \pm 15\%$ relative humidity in a controlled light/dark cycle (light on between 8:00 a.m. and 8:00 p.m.). Animals had free access to food and water except during manipulations and behavioral assessment. Experiments were performed blinded for NTG, genotype or pharmacological treatment depending on the studied condition.

### Model of chronic migraine
Animals were exposed to a schedule of repeated NTG injections previously used to precipitate long lasting mechanical hypersensitivity[6]. Briefly, mice were injected with 10 mg/kg NTG or its vehicle every other day for 8 days (5 i.p. injections total). Two different NTG formulations were tested: first, 5 mg/1.5 ml ampoules and second, 50 mg/50 ml vials (Bioindustria LIM, Novi Liguri, Italy). The 5 mg/1.5 ml ampoules contained NTG dissolved on a vehicle made of 1 ml propylene glycol and 0.5 ml ethanol (Bioindustria LIM). Such solution was further dissolved in saline to obtain 1 mg/ml NTG, reaching final concentrations of 10% ethanol and 20% propylene glycol. This preparation was no longer used after the first behavioral experiment because the treatment with vehicle induced mechanical nociceptive sensitization by itself as shown in Supplementary Fig. 1. The second NTG formulation consisted of 50 mg/50 ml NTG vials containing a vehicle made of 5% dextrose and 0.105% propylene glycol in pure water (Bioindustria LIM). This NTG formulation or its vehicle was administered without further dilution. Such preparation was chosen to complete all the remaining experiments because the vehicle did not modify nociceptive sensitivity. Both the first and the second preparations were administered at 10 ml/kg intraperitoneally (i.p.), obtaining the same NTG dose of 10 mg/kg as described[6].

### Experimental design
Detailed description of the rationale, timing and experimental design followed in each experiment can be found in Supplementary Methods within the Supplementary Information file.

### Assessment of mechanical sensitivity
Mechanical thresholds were quantified by measuring the hind paw withdrawal response to von Frey filament stimulation. To decrease stress, prior to baseline measurements mice were habituated for 4 h to the testing environment during 2 days. On the evaluation days, animals were also allowed to habituate for 1–2 h before testing in order to obtain appropriate behavioral immobility. Briefly, animals were placed in Plexiglas® chambers ($10 \times 10 \times 14$ cm) with a wire grid bottom through which the von Frey filaments (bending force range from 0.008 to 2 g) (PanLab, Cornellá, Barcelona, Spain) were applied, by using the up-down paradigm as described[75]. The filament of 0.4 g was first applied. Then, the strength of the next filament was decreased when the animal responded or increased when the animal did not respond. The upper limit value (2 g) was recorded as a positive response even if there was no withdrawal response, and the lower limit was recorded as negative even if there was withdrawal response (0.008). This up-down procedure was stopped 4 measures after the first change in animal responding (i.e., from response to no response or from no response to response). The sequence of the last 6 responses was used to calculate the mechanical threshold. Both ipsilateral and contralateral hind paws were alternatively tested whenever possible, and stimuli were applied at a minimum of 2 min intervals to avoid hypervigilance or sensitization between successive filament applications. Filaments were completely bent before considering responses and hold up to 4–5 s to consider a negative response. Clear paw withdrawal, shaking or licking were considered as nociceptive-like responses. The responses of both hind paws were averaged to obtain the mechanical threshold of each individual.

### RNA extraction and RT-PCR
Trigeminal ganglia were isolated and frozen on dry ice until RNA extraction. Tissue was homogenized on ice using a polytron (Polytron PT 2000 Kinematica AG, Malters, Switzerland) and RNA was extracted with a TRIZOL (15596-026 Thermo Fisher Scientific) extraction method as described[76]. Briefly, chloroform was added to yield 2 phases, including one hydrophilic phase at the top containing RNA. Then after centrifugation at $12,000 \times g$, 10 min at 4 °C the aqueous phase was mixed with isopropanol to precipitate RNA. The resulting pellet was washed with ethanol and air-dried. RNA purity and quantity were assessed by spectrophotometry (NanoDrop 2000, Thermo Fisher Scientific). Retrotranscription to cDNA was conducted with First Strand Synthesis Kit (K1612, Thermo Fisher Scientific) using dT primers. Primers for cDNA amplification were: TRPA1 fw 5′-GCAGGTGG AACTTCATACCAA and rv 5′-CACTTTGCGTAAGTACCAGACTGG, TRPM8 fw 5′-CTTTCTAAGCAATGGTATGGAG and rv 5′-GGTTTCTTCC TAAATGATACGAG, GAPDH fw 5′-CCAATGTGTCCGTCGTGGATCT and rv 5′-GTTGAAGTCGCAG GAGACAACC. Relative expression values were obtained by applying the equation $2^{-(\Delta CT\ gene\ of\ interest - \Delta CT\ GAPDH)}$.

### Mouse trigeminal primary cultures
Adult mice (8–14 weeks) were sacrificed by cervical dislocation and trigeminal ganglia were extracted, micro-dissected from nerve projections and kept in ice-cold HBSS solution. Trigeminal ganglia were incubated in collagenase 48 µg/ml, 3.5 U/mg (C7657, Merck) and dispase 3 mg/ml, 1.79 U/mg (17105-041, Thermo Fisher Scientific, Waltham, USA) for 45 min at 37 °C in 5% $CO_2$[77]. Thereafter, samples were mechanically disaggregated by pipetting 15 times through a 1 ml blue plastic tip. Neurons were separated from other cell types and tissue debris by placing the tissue homogenate over a 15% BSA solution (073k7601, Merck) and centrifuged 7 min at $300 \times g$[78]. Then, cells were

seeded in 5 µl drops over crystals treated with 8.3 µg/ml poly-L-Lysine (P9155, Merck) and 5 µg/ml laminin (L2020, Merck). In total, 30 min after seeding cultures were supplemented with 500 µl of culture medium and incubated for 12–16 h at 37 °C and 5% $CO_2$[77]. Cells were kept in a hormone-free culture medium consisting of Dulbecco's Modified Eagle Medium/F12 (DMEM/F12) without Phenol Red (11039021, Thermo Fisher Scientific), 1X Minimum Essential Medium (MEM) vitamin solution (11120052, Thermo Fisher Scientific), 1% penicillin/streptomycin (15140-022, Thermo Fisher Scientific) and home-made µg/ml $N_2$-containing insulin (I2643, Merck), 0.1 mM putrescine hydrochloride (P7505, Merck), 3 nM sodium selenite (S5261 Merck), 100 µg/ml transferrin (T2872, Molecular Probes, Eugene, USA), 25 ng/ml NGF (G5141, Promega, Madison, USA) and 25 ng/ml hGDNF (450-10, Peprotech, London, UK).

### Fluorescence Ca²⁺ imaging

Non-ratiometric calcium imaging experiments were conducted with the fluorescent indicator fluo4-AM (F14201, Thermo Fisher Scientific). Trigeminal neurons or Human Embrionic Kidney 293 (HEK293) cells (CRL-1573 ATCC) were incubated with 5 mM (6 mg/ml) fluo4-AM and 0.2% w/v pluronic acid (F-127, Thermo Fisher Scientific) for 60 min at 37 °C in standard extracellular solution (in mM: 140 NaCl, 3 KCl, 2.4 $CaCl_2$, 1.3 $MgCl_2$, 10 HEPES, and 5 glucose, adjusted to pH 7.4 with 1 M NaOH)[79]. Afterwards, cells were washed with standard extracellular solution for at least 20 min. Fluorescence measurements were obtained on an inverted microscope (Axiovert 200/B, ZEISS) coupled to a Hamamatsu FLASH 4.0 LT camera (C11440-42U30, Hamamatsu, Sunayama-cho, Japan). Before starting the experiment, an image of the microscopic field was obtained with transmitted light to create the regions of interest around cells with neuronal morphology. Then, Fluo4 was excited at 480 nm (excitation time 200 ms) with a rapid gating shutter (lambda-shutter 10/2 Sutter instruments, Novato, USA). Mean fluorescence intensity was recorded for each cell with HCimage DIA software (Hamamatsu Photonics) every 3 s. Calcium imaging recordings were performed at 35 °C. Response sizes after agonists were calculated by measuring peak minus basal values and divided by the positive control of the experiment, i.e., 40 mM KCl in trigeminal primary cultures, and 10 µM ionomycin in HEK293 and IMR90 human cell cultures. Responses were scored as positive if the increase in fluorescence was >0.2 arbitrary units[79]. Substances dissolved in extracellular solution were delivered through a high-flow rate perfusion system controlled with an automatic system of valve clamps (PC-16 Bioscience Tools, S. Diego, USA). Cells were washed with extracellular solution between calcium responses for a period of at least 300 s to ensure recovery of basal fluorescence levels. Example dataset of calcium imaging (Fig. 2c, d) and the analysis instructions have been deposited in Github: https://github.com/greferball/Ca_Imaging_NatComm_2022.git

### Drugs tested in behavioral studies

The TRPM8 selective blocker AMTB hydrochloride (AMTB, N-(3-aminopropyl)−2-{[(3-methylphenyl) methyl] oxy}-N-(2-thienylmethyl) benzamide hydrochloride, Tocris, Bristol, UK) was dissolved in dimethyl sulfoxide (DMSO, Merck, Darmstadt, Germany) and was further diluted in saline to reach 2.5% DMSO. The range of AMTB doses (5–15 mg/kg i.p., 10 ml/kg) was chosen after dose conversion[80] from previous experiments demonstrating efficacy in attenuating hyperactive bladder activity in rats[81]. Mechanical sensitivity was measured 30 min after AMTB, based in previous behavioral results in mice[82].

The potent and selective TRPM8 agonist WS12 ((1R,2S,5R)−2-Isopropyl-N-(4-methoxyphenyl)−5-methylcyclohexanecarboxamide, Tocris) was dissolved in DMSO and diluted in corn oil to reach 2.5% DMSO. The range of WS12 i.p. doses (5–15 mg/kg i.p., 10 ml/kg) was chosen based on previous data showing

antinociceptive effects of WS12 in mice[33]. WS12 was administered 30 min prior assessment of mechanical sensitivity[33]. In the formalin test, WS12 was dissolved in DMSO and diluted in saline up to 0.6% DMSO to achieve an amount of 6 nmol in 20 µl as described[33]. We used WS12 and not menthol or icilin as a TRPM8 agonist to avoid unspecific signaling over TRPA1[83].

Testosterone (T1500, Merck) was dissolved in 45% 2-Hydroxypropyl-β-cyclodextrin in water[84] to obtain a solution of 22 mg/ml for sustained delivery. This solution or its vehicle was introduced inside Alzet osmotic minipumps (Model 2004, Durect Corporation, Cupertino, CA, USA) to achieve a testosterone delivery of 6 µg/h, based on previous studies obtaining significant effects in orchidectomized mice[29,85]. In the experiments assessing the acute effect of testosterone in females, testosterone was dissolved in DMSO and diluted in corn oil to reach 2.5% DMSO as with WS12, and it was administered at 1 mg/kg, 1 h before the assessment of mechanosensitivity based on previous data assessing its effects on anxiety-like behavior[86].

### Drugs for cellular studies

Fluo4-AM (F14201, Molecular Probes) was dissolved in DMSO at a concentration of 2 µg/µl used on HBSS at a concentration of 6 µg/µl to load the cells for calcium experiments. In calcium experiments, cells were under a constant flow of a standard extracellular solution (in mM: 140 NaCl, 3 KCl, 2.4 $CaCl_2$, 1.3 $MgCl_2$, 10 HEPES, and 5 glucose, adjusted to pH 7.4 with 1 M NaOH)[79], and stimuli were applied dissolved in this solution. Substances used in calcium imaging were 100 µM AITC (W203408, Merck) dissolved in 0.001% DMSO, 500 nM WS12 (3040/50, Tocris) dissolved in 0.01% DMSO, 10 pM testosterone dissolved in 0.01% DMSO, 10 µM ionomycin (I9657 Merck). NTG at 100 µM (Biondustria LIM) was used in calcium imaging and in the CGRP release assay in the formulation containing 5% dextrose and 0.105% propylene glycol as vehicle. In the CGRP release experiment, 10 µM DD04107 (BCN Peptides SA, San Quintí de Mediona, Spain) dissolved in water was used[79].

### CGRP release assay

Trigeminal cultures were incubated for 48 h, then cells were treated with the exocytosis inhibitor 10 µM DD04107 (BCN Peptides SA, San Quintí de Mediona, Spain) or its vehicle ($H_2O$) for 1 h[77]. Next, cells were treated with 100 µM NTG or its vehicle (5% dextrose and 0.105% propylene glycol) for 30 additional min. Incubation solutions were made in culture medium and kept at 37 °C and 5% $CO_2$.

### Immunocytochemistry

In total, 30 min after NTG exposure (vehicle 5% dextrose and 0.105% propylene glycol), the media was removed from the cells and the culture was washed with 1X PBS (D8662, Merck) 3 times[79]. Afterwards, cells were fixed with 4% paraformaldehyde (158127, Merck) for 20 min at room temperature. Permeabilization was achieved with 0.1% v/v Triton 100X (P8787, Merck) for 5 min and blocking with 5% Normal Goat Serum (NGS, G9023, Merck) for 1 h, both in 1X PBS. Neurons were labeled with primary antibodies rabbit anti-MAP 1:250 (17490-1-AP, LabClinics, Barcelona, Spain) and mouse anti-CGRP 1:200 (AB81887, Abcam, Cambridge, UK) incubated overnight at 4 °C. Secondary antibodies goat anti-rabbit Alexa 488 1:1000 (A11034, Thermo Fisher Scientific) and goat anti-mouse Alexa 568 1:1000 (A11031, Thermo Fisher Scientific) were incubated for 1 h at room temperature and protected from light. Nuclei were stained with DAPI 1.5:10000 (D9564, Merck) for 5 min at room temperature. Slides where mounted with mowiol (475904, Merck) and images taken with a confocal microscope (LSM 900, ZEISS, Jena, Germany). Mean fluorescence intensity for each cell was obtained with ImageJ (Wayne Rasband, NIH), and the average value of positive cells was calculated for each picture.

## Culture and transfection of human cell lines

Human embryonic kidney 293 cells (HEK293) were maintained in DMEM plus Glutamax, supplemented with 10% Fetal Bovine Serum (FBS, Thermo Fisher Scientific) and 1% penicillin/streptomycin, and incubated at 37 °C in 5% $CO_2$ atmosphere. For the study of TRPA1 function, HEK293 cells were plated in 24-well dishes at $2 \times 10^5$ cells/well and transiently transfected with human TRPA1 in a pCMV6-AC-GFP vector (Viktorie Vlachova, Czech Academy of Sciences) using Lipofectamine 3000 (Thermo Fisher Scientific)[87]. For the transfection, 2 ml of Lipofectamine 3000 was mixed with the DNA in DMEM plus Glutamax with 1% FBS[87]. IMR90 fibroblast-like cells (CCL-186 ATCC, Virgina, USA) were seeded in 12 mm coverslips at 50,000 cells/well and were maintained in MEM enriched with 10% FBS and 1% penicillin/streptomycin at 37 °C in 5% $CO_2$ atmosphere[88]. For the initial experiments assessing TRPM8 function, HEK293 cells constitutively expressing human or rat TRPM8[35] were obtained from Prof. Belmonte Laboratory (Instituto de Neurociencias, San Juan, Alicante, Spain). All calcium imaging and electrophysiological recordings took place when cells reached 60% of confluence and 72 h after transfection when applicable.

## Chemically-induced nocifensive behavior (formalin test)

Mice were individually placed into transparent chambers and were habituated for 1 h before testing. To evaluate the pain-relieving effect of WS12 against formalin-induced nocifensive behavior, 20 μl of a 45% 2-Hydroxypropyl-β-cyclodextrin solution containing 5% formalin (F8775, Merck) and 0.6% DMSO with or without 6 nmol of WS12 were injected subcutaneously into the plantar aspect of the right hind paw (i.pl.) by using a Hamilton syringe (Hamilton Syringe Gastight™ serie 1700, TLL end, Merck) coupled to a 30-gauge needle. The time spent expressing nocifensive behavior (licking or biting of injected paw) was quantified in 5 min intervals during 60 min as described[34]. To assess the influence of endogenous TRPM8 activity on the extinction of nocifensive behavior, mice received first 20 μl of 5% formalin dissolved in saline in their right hind paw, and nocifensive behavior was quantified as above. After the initial 60 min when the nocifensive responses ceased, AMTB (10 mg/kg) or vehicle (2.5% DMSO in saline) were administered i.p. and behavioral quantification continued for 1 h.

## Orchidectomy

Orchidectomy was adapted from a previous procedure[89] changing to a single midline scrotal incision to minimize tissue injury. Briefly, mice were anesthetized with a mixture of i.p. ketamine (75 mg/kg; Imalgene, 100 mg/ml, Boehringer Ingelheim, Ingelheim/Rhein, Germany) and xylazine (15 mg/kg, Merck) and a single midline scrotal incision was made. The testes were exposed, and the vas deferens and testicular blood vessels were ligated with 2 tight knots of 6–0 black silk (8065195601, Alcon Cusi S.A., Barcelona, Spain). An incision was made between the 2 knots to remove testes and epididymis and the incision was closed with three additional square knots after ensuring hemostasis. Sham surgeries were performed similarly but the testicles were exposed and not ligated or removed. Subsequent nociceptive evaluations were conducted 3 weeks after surgeries.

## Testosterone replacement treatment

Testosterone or its vehicle (45% 2-Hydroxypropyl)-β-cyclodextrin in water) were placed into Alzet osmotic minipumps (Model 2004, 0.23 μl/h for 28 days) following manufacturer instructions. Minipumps were implanted subcutaneously between the scapulae under ketamine (75 mg/kg)–xylazine (15 mg/kg) anaesthesia. The pump was set to deliver vehicle or testosterone at an estimated dose of 6 μg/h, based on previous studies obtaining significant effects in orchidectomized mice[29,85]. Testing and NTG injections began after 3 days of minipump implantation.

## Patch-clamp whole-cell recordings

Whole-cell recordings were conducted using an EPC-10 amplifier with Patchmaster software (both from HEKA Electronik, Dr. Schulze GmbH, Germany). Patch pipettes, prepared from thin-wall borosilicate capillary glass tubing (Warner Instruments, Hamden CT, USA) were pulled with a horizontal flaming/brown Micropipette puller Model P-97 from Sutter Instruments to a final resistance of 2–4 MΩ when filled with internal solution. Recordings were acquired at 10 kHz and low-pass filtered at 3 kHz. Recordings with leak currents >200 pA or series resistance >15 MΩ were discarded. Cells were seeded on 12 mm diameter glass coverslips treated with poly-L-lysine solution. Intracellular pipette solution contained in mM: 150 NaCl, 3 $MgCl_2$, 5 EGTA and 10 HEPES, pH 7.2 with CsOH. Extracellular physiological solution contained (in mM): 150 NaCl, 6 CsCl, 1 $MgCl_2$, 1.5 $CaCl_2$, 10 glucose and 10 HEPES, pH 7.4 with NaOH. All measurements were performed at room temperature. After seal opening, cells were maintained at a holding potential of −60 mV for few minutes and then a voltage ramp from −120 to +120 mV of 300 ms duration was applied.

## Knockdown of androgen receptor in HEK 293 cells expressing human TRPM8

HEK293 cells expressing human TRPM8[35] (Prof. Belmonte Laboratory, Instituto de Neurociencias, San Juan, Alicante, Spain) were cultured in EMEM (ATCC, 30-2003), 1% penicillin/streptomycin (Gibco, 15140-122), 1% geneticin (Gibco, 10131-019), 10% FBS (Gibco, 16000-044), 2 mM L-glutamine (Gibco, A2916801) and 2.5 mg/ml glucose (Sigma, G8270). T25 flasks at 80% confluency were transfected with DMEM plus Glutamax (Gibco, 61965-026), 1% Penicillin/Streptomycin (Gibco, 15140-122), 19.6 μl of Lipofectamine 3000 (Invitrogen, L3000-015), 1 μg of androgen receptor siRNA (Thermo Fisher, Ambion, s1538; sense GCCCAUUGACUAUUACUUUtt; antisense AAAGUAAUAGUCAAUGG GCaa) or negative control scramble siRNA (Thermo Fisher, Ambion, 4390843) and 6.5 μg of pcDNA3eGFP plasmid DNA (Addgene plasmid #13031). After 72 h, GFP expression was assessed to ensure that the transfection had been successful and western blot or patch clamp were performed. GFP was excited at 500 nm with an exposure time of 500 ms and emitted fluorescence was filtered at 535 nm (Lambda-10-2-filter wheel, Sutter Instruments). The cells were imaged under a ×10 air objective (Axiovert 200 inverted microscope, Carl Zeiss) with an ORCA Flash LT camera (Hamamatsu Photonics) and visually compared to non-transfected hTRPM8 HEK cells to estimate the efficiency of transfection. For patch clamp experiments, cells were seeded in 12 mm poly-L-lysine (Sigma-Aldrich, P4707) treated coverslips at 20,000 cells/well.

## Protein extraction and western blotting

Total protein from HEK 293 cells was extracted using lysis buffer (150 mM NaCl, 50 mM Tris-Base, 1 mM EDTA, 1% Triton X-100, 0.5% sodium deoxycholate, 0.1% SDS, pH 8) containing 1:100 Halt Protease Inhibitor Cocktail EDTA-free (87785, Thermo Scientific). After 40 min incubation with lysis buffer, solubilized proteins were collected by centrifugation at $9400 \times g$ for 15 min. Protein concentration was determined with Pierce BCA Protein Assay Kit (23225, Thermo Scientific). In total, 20 μg of total protein were prepared in loading buffer (10X: 600 mM Tris-HCl, 400 mM DTT, 1% bromophenol blue, 20% SDS, 50% glycerol, pH 6.8) and heated up to 95 °C for 5 min before loading the gel. Proteins were separated by SDS-PAGE in 7.5% gels. Electrophoresis ran in running buffer (0.3% Tris-Base, 1.44% glycine, 0.1% SDS, pH 8.4) at 100 V during the first 10 min, and then at 150 V until the end. Separated proteins were transferred onto a 0.45 μm nitrocellulose membrane (Bio-Rad) using a wet blotting system for 2 h at 100 V and 4 °C in transfer buffer (25 mM Tris-Base, 190 mM glycine, 0.1% SDS, 20% methanol, pH 8.4). Afterwards, the membrane was blocked with 5% BSA for 1 h. Androgen receptor (AR) and β-tubulin were probed with

specific primary antibodies anti-AR (1:100 in 1% BSA/TBST; sc-7305, Santa Cruz Biotechnology) and anti-β-tubulin (1:1000 in 2.5% BSA/TBST; 10094-1-AP, Proteintech) and were incubated overnight (16 h) at 4 °C. After six 5-min washes with TBST (20 mM Tris-Base, 150 mM NaCl, 0.1% Tween-20, pH 7.5) the membrane was incubated with secondary antibodies anti-mouse IgG-HRP (1:10,000 in 1% BSA/TBST; A4416, Sigma) or anti-rabbit IgG-HRP (1:25,000 in 2.5% BSA-TBST; A0545, Sigma) for 1 h at room temperature. After six 5-min washes with TBST, proteins were detected with substrate SuperSignal West Pico Plus (34577, Thermo Scientific). Chemiluminescence was visualized with ChemiDoc imaging system (Bio-Rad).

### Statistical analyses

Time courses of nociceptive behavioral data conducted in male and female mice were analyzed using two-way repeated measures ANOVA with time as within-subjects factor and NTG treatment or genotype as between-subject factors. The time courses involving orchidectomized animals were analyzed with three-way repeated measures ANOVA, with time as within-subjects factor and either NTG and orchidectomy or genotype and testosterone as between-subject factors. Levene's test of equality of error variances and Mauchly's sphericity tests were used to assess normality of the data and Bonferroni post hoc pairwise comparisons were subsequently conducted when appropriate. Three-way ANOVA was also used to analyze the data of WS12 experiments (time point, WS12, NTG) whereas a within-design was chosen to analyze the effects of the AMTB doses in wild-type males recovered from sensitization (Friedman's test followed by Benjamini adjustment). A three-way ANOVA was also used to analyze AMTB effects on NTG-exposed orchiectomized animals (Time point, Testosterone, Genotype). The time-course data of the chemically-induced nocifensive behavior was analyzed with repeated unadjusted $t$-tests to avoid assumptions of similar variances for the first and the second phases of the formalin test and posterior measurements. For the cellular studies, data normality was first assessed with D'Agostino–Pearson or Shapiro–Wilk test for the patch recordings. Then, comparisons of two groups were analyzed accordingly with $t$-tests or Mann–Whitney $U$ tests, whereas comparisons of more than two groups were analyzed with either one-way ANOVA followed by Bonferroni or Kruskal–Wallis followed by Mann–Whitney $U$ tests. RT-PCR and cellular data containing two factors were analyzed with two-way ANOVA followed by Bonferroni. Sample size for animal and cellular studies was estimated based on previous experience in our laboratories using similar behavioral and cellular approaches[79,87,90]. Raw data can be found in the Source Data File and results of the statistical tests are included in the Supplementary Information file. Throughout the manuscript, data are expressed as mean ± SEM values and individual datapoints are shown whenever possible. Differences were considered statistically significant when $p$ values were <0.05. Statistical analyses were performed using IBM SPSS 25 Software (IBM, Chicago, IL, USA), Microsoft Excel 2019 MSO (Redmond, WA, USA) and GraphPad Prism 7.04 (GraphPad Software Inc., San Diego, CA, USA). Additional distribution data can be found in the Supplementary Statistical Results subsection of the Supplementary Information file.

### Reporting summary

Further information on research design is available in the Nature Research Reporting Summary linked to this article.

## Data availability

All source data from the figures are available within the manuscript as a Source data file and results of statistical analyses are provided within the Supplementary Information file. Example dataset of calcium imaging (Fig. 2c, d) and analysis instructions have been deposited in Github: https://github.com/greferball/Ca_Imaging_NatComm_2022.git. All the calcium imaging datasets generated and analyzed during the study are available from the corresponding authors on request. Data used for computational analyses are available as Supplementary Information in the zip folder Computational_files.zip which contains instructions in a readme.txt and scripts. Source data are provided with this paper.

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

## Acknowledgements

We are thankful to the following funding bodies: grants from MICIN/AEI no. SAF2015-66275-C2-1-R and RTI2018-097189-B-C21 (DOI /10.13039/501100011033, ERDF A way of making Europe, by the "European Union") to A.F.M. and A.F.C., and fellowship grants BES-2016-077985 to D.A.A. and PRE2019-091317 to J.d.A.L.; grants form GVA, PROMETEO/2021/031 to A.F.M. and A.F.C.; UMH-PAR2019 to A.F.C. and A.F.M.; GVA fellowship grant GRISOLIAP/2019/09444/19 to S.G. Authors thank Gema Osuna Tenorio, Laura Butrón and José Manuel Serrano García for their help and technical expertise. We thank Dr. F. Viana for providing the TRPM8 knockout mice.

## Author contributions

D.A.A. conducted calcium imaging, cell culture, immuno-fluorescence, RT-PCR experiments, performed formalin behavioral assays, designed experiments, analyzed the data, wrote and edited the first and subsequent drafts of the manuscript. D.C. conducted all behavioral assays, performed surgeries, analyzed the data, conceptualized and designed experiments, coordinated in vivo, in vitro and computerized experiments and wrote and edited the first and subsequent drafts of the manuscript. J.d.A.L. executed and analyzed docking experiments, conducted protein extraction, western blotting and cell cultures and wrote the first and subsequent drafts of the manuscript. M.N.K. conducted the electrophysiological assays and wrote manuscript. S.G. conducted knockdown experiments and wrote manuscript. G.F.B. supervised and executed docking experiments and revised and edited the manuscript. A.F.C. supervised and designed experiments, conceptualized the project, revised and edited the manuscript and provided funding. A.F.M. supervised and designed experiments, conceptualized the project, wrote, revised and edited the manuscript and provided funding.

## Competing interests

The authors declare no competing interests.

## Additional information

**Correspondence and requests** for materials should be addressed to David Cabañero, Asia Fernández-Carvajal or Antonio Ferrer-Montiel.

