## [Peer Review File · Nature Communications]

TRPM8 contributes to sex dimorphism by promoting recovery of normal sensitivity in a mouse model of chronic migraineREVIEWER COMMENTS

Reviewer #1 (Remarks to the Author):

In this manuscript the Authors investigated the involvement of TRPA1 and TRPM8 in migraine pain and sexual dimorphism by using chronic and intermittent treatment with nitroglycerin (NTG) as a pre-clinical model of migraine. The key findings show that TRPM8, but not TRPA1, channels are involved in the sexual dimorphism observed in mechanical hypersensitivity induced by chronic NTG injection. It is suggested that the antinociceptive resilience observed in male mice is due to the effect of testosterone; this latter is suggested to activate TRPM8 channels, driving the recovery of normal sensitivity in males.

The present study provides interesting insight and suggestions for further analysis on migraine pain and sexual dimorphism. A considerable amount of data has been performed by the Authors; however, the experimental design and consequently the findings and discussion, appear to be not properly organized. This makes the manuscript uneasy to follow by the reader even it is written in a comprehensible way. Below are reported some comments that need to be addressed by the Authors prior to publication.

- 1)The abstract should be revised, since it is not clearly stated the aim of the study and the results are confused. It is unclear why TRPA1 channels have been investigated: the link with TRPM8 is missing. In line 42 the Authors refer to "preclinical models of acute and chronic pain..."; to define such models.
- 2)The state of art section could be improved. The aim of the study is not defined; this must be addressed. In light of this issue, to consider a possible revision of the title.
- 3)The material and methods section is quite chaotic and difficult to follow. Bibliographic references of most of the dosage/timing/methods are missing (sections 2.2., 2.5., 2.6., 2.7., 2.8., 2.9., 2.12). Please provide where appropriate
- 4)Please revise the material and methods section to better describe the experimental groups/sets and the specific number of animals used for each evaluation (NTG model and mechanical sensitivity, trigeminal cultured cells to CGRP assay and IHC, ect). The experimental design is not clear: a brief description of the timing of each assessment would be useful to better follow all experimental evaluations; tables and/or schemes may be of help.
- 5)Section 2.2: to explain why two formulations of NTG were used.
- 6)Why did the Authors opt for the evaluation of mechanical allodynia in the hind paw and not in the trigeminal area?
- 7)Why did the Authors perform plantar formalin test instead of the orofacial version? And more importantly, why didn't they combine the formalin test with the NTG model?
- 8)Section 2.11: to include timing and dosage of TRPM8 agonist and antagonist. To motivate why the TRPM8 agonist was given before and the antagonist after formalin injection.
- 9)Section 2.12:to provide a description of drugs treatment (NTG, TRPM8 antagonist, etc) of the orchidectomized mice.
- 10)At what cycle phase were female mice evaluated for behavioral outputs? Did the authors use vaginal cytology to verify the estrous cycle stage?
- 11)Section 2.15: the sample size calculation and distribution data should be included.
- 12)The timing of Figure 1 is confusing. Please use consistency for days' numbers (panel A vs plots and figure legend).
- 13)All figures showing NTG-induced allodynia in males and females' mice (acute and chronic NTG) should be displayed in a single graph.
- 14)The organization of results are quite confusing, it is lacking a logical sequence of the data presented. Moreover, it is not clear why in some circumstances the same evaluations were performed only in females or males. For instance, AMTB within the formalin test was only tested in male mice.
- 15)The discussion section is also a bit confused. It is not clear whether the involvement of TRPM8 is related to its activation or desensitization. By using a TRPM8 agonist the Authors found no effect in female mice, thus it is unclear in which way this channel may operate in the prolonged chronic sensitization induced by NTG.

Reviewer #2 (Remarks to the Author):

This is an extremely interesting study linking the resistance of male mice to chronic mechanical hypersensitivity to the agonist action of testosterone on the cation channel TRPM8. The results shed light on the mechanisms underlying the sexual dimorphism of chronic migraine are therefore of wide interest. I have a number of comments that may serve to improve the quality of the manuscript.

1.- As a matter of style, I suggest not starting the Results with the effects of the vehicle that produces mechanical hypersensitivity. This is interesting, but it is just a side detail. I would propose to just make a small comment about this. In fact, the effects of the vehicles should be presented after describing the administration protocol (We injected 10 mg/kg NTG or vehicle...).

2.- Line 405, the term "calcium imaging activity" is not correct.

3.- In my opinion, the results of the intracellular Ca²⁺ imaging experiments are difficult to interpret. Ideally, direct (ratiometric) measurements of intracellular Ca²⁺ should have been performed to be able to compare the amplitude of signals elicited in neurons from different animals. Differences may be attributed to distinct loading of the dye.

It is unclear whether the normalization to the amplitude of the responses to high K⁺ is appropriate. For this to work, one has to assume that the treatments do not alter the expression and function of voltage-gated Ca²⁺ channels, the resting potential or several other factors that determine the response to high K⁺.

It is unclear how the statistical analysis on Ca²⁺ imaging was performed. What are exactly the values presented in Fig. 2E, 2G, 3E and 4F? How many neurons were measured in each condition? The legend states that the data points represent different experiments, does this mean different mice?

It is stated that the responses to AITC are of longer duration in neurons of NTG-treated animals, but no statistics are given to back up this claim.

I would like to indicate that none of these elements are crucially essential for the interpretations of the main results of this study, but it would be nice that they are considered to improve the quality of the manuscript.

4.- Fig. 2D should show also the responses to high K⁺.

5.- In Fig. 2F, the green traces are not clearly distinguishable from the black traces, please use a better color.

6.- Lines 427-428, it is unclear what is meant by "...revealing that NTG sensitize through the activation of TRPA1".

7.- Lines 467-468, it is stated that WS12 elicited Ca²⁺ transients with similar morphologies..., but no supporting data is provided. Visual inspection of the traces is not sufficient.

8.- Line 563, please delete "novel". What is novel is the discovery, not the role of TRPM8. Please, also note that this sentence is too long.

9.- Line 567, the statement "Testosterone by activating TRPM8..." and the following sentence are too strong. I strongly suggest to write "Our results suggest that...", or "We propose that...".

10.- Line 576, a period is missing before "Overall...".

11.- Line 581, please delete the allusion to TRPA1 as thermoTRP. This may introduce an unnecessary element of confusion in the context of the present study.

12.- Line 634, please write "AMTB".

13.- Line 673, it is unclear what "TRPM8 testosterone activity" means. Testosterone-induced TRPM8 activation? Same for "calcium activity" in line 679.

14.- The docking simulations should belong to the Results section.

15.- Lines 693-694, it is unclear to what "Additional less popular docking solutions" is referring to.

16.- How dose activation of TRPM8 result in reduced mechanical hypersensitivity?

17.- Can the authors make any "biological sense" of the sexual dimorphism of chronic migraine? Are females just doomed to suffer from chronic pain by their lack of testosterone?

18.- Do the authors expect that their results can be extended to other TRPA1-mediated chronic pains?

Reviewer #3 (Remarks to the Author):

Using mouse behavior and calcium imaging experiments, the authors put forth the concept that women experience more severe migraines than men because testosterone binding to TRPM8 has a pain-relieving effect. This work has clearly demonstrated that the presence of TRPA1 contributes to hypersensitivity and the presence of TRPM8 contributes to reduced hypersensitivity in mice that have been administered nitroglycerin – intended to model chronic migraine. It would be nice to see further validation of this murine migraine model. For instance, instead of measuring solely mechanical sensitivity, one could also measure cortical spreading depression. It is an interesting finding that both TRPA1 and TRPM8 play a role in mechanical sensitivity in this mouse model, and by extension possibly migraine pain in humans. Furthermore, there does appear to be sex differences, something that is also worth exploring. However, the claim put forth that androgenic TRPM8 activity drives sexual dimorphism of chronic migraine has not been substantiated. Given the controversy surrounding whether testosterone is a TRPM8 agonist, additional experiments are required to demonstrate that testosterone binds directly to TRPM8 (and that the effects are not indirect). Knock-out mice and AMTB inhibition show that TRPM8 is linked to the behavioural phenotype, but not that TRPM8 is a testosterone receptor. While calcium imaging experiments have been conducted (with modest effect), electrophysiology would be far more convincing. Computational docking studies based on homology models of a 4.1 Å structure are not suitable to demonstrate ligand binding, especially given that this pocket has been shown to be basically nondiscriminatory. It should be noted that both progesterone and estradiol were also able to be positioned in the cavernous and adaptable ligand-binding pocket of TRPM8. As testosterone's primary mechanism of action is through binding to the androgen receptor, should androgen receptor knock-out mice not also be evaluated? The scientific foundation of this study (that TRPM8 is a testosterone receptor) is not solid enough to allow for the numerous assumptions that have been made in interpreting the data. Assembling this data into a manuscript with a different focal point is suggested.

Matthijs Dorst, Ph.D.
Associate Editor
Nature Communications

Reference: **NCOMMS-21-39875A**

Dear Dr. Dorst:

Enclosed please find an amended version for the manuscript entitled **Androgenic TRPM8 activity drives sexual dimorphism in a murine model of chronic migraine** by David Alarcón-Alarcón, David Cabañero, Jorge de Andrés-López, Magdalena Nikolaeva, Simona Giorgi, Gregorio Fernández-Ballester, Asia Fernández-Carvajal and Antonio Ferrer-Montiel.

We thank you for the opportunity to address the concerns raised by the reviewers and submit an amended version. We have addressed them and performed additional experiments as suggested. Patch clamp experiments conducted in HEK293 cells expressing TRPM8 revealed testosterone-induced currents that were insensitive to the knockdown of the androgen receptor using a specific siRNA. These currents are abolished with the selective TRPM8 antagonist AMTB and are not found in naïve HEK293 cells void of heterologous TRPM8 receptor expression. We have incorporated these data into the manuscript, as indicated. In addition, we also discussed on the recent paper by McKemy's group et al. published in PAIN (March 29th, PMID: 35353773) reporting apparent diverging results on TRPM8 contribution to the mechanical sensitization in the chronic migraine model (Discussion, page 20, middle of 2nd paragraph). We believe that all suggestions have pivotally contributed to increase the quality of our study and are grateful to the referees for their constructive comments.

Dr. Magdalena Nikolaeva-Koleva and Simona Giorgi were added as authors since they were essential for the electrophysiological studies and the experiments knocking down the androgen receptor.

We trust you find this amended manuscript appropriate for publication in Nature Communications. We look forward to your prompt and positive reply.

David Cabañero
Asia Fernández-Carvajal
Antonio Ferrer-Montiel

We turn now to address the specific concerns raised by the reviewers. Changes in the manuscript are highlighted in blue.

REVIEWER COMMENTS

Reviewer #1(Remarks to the Author):

1)The abstract should be revised, since it is not clearly stated the aim of the study and the results are confused. It is unclear why TRPA1 channels have been investigated: the link with TRPM8 is missing. In line 42 the Authors refer to “preclinical models of acute and chronic pain...”; to define such models.

The abstract (page 2) was revised to clarify the aim of the study. The reasons for studying TRPA1 and TRPM8 were explicitly included:

“The mechanisms contributing to the high prevalence of chronic migraine in females are yet elusive. A trigger of this sexual dimorphism may be found in TRPA1 and TRPM8, two transient receptor potential channels expressed in trigeminal neurons that were related

to migraine pathophysiology. To investigate their involvement, we used a mouse model of nitroglycerine-induced chronic migraine that displays a sexual dimorphic phenotype, characterized by persistent mechanical hypersensitivity solely in females, as males readily recover from the migraine crisis.”

The models of acute and chronic pain were defined and the results were organized to increase clarity, including the new results:

“Experiments in constitutive knockouts revealed an essential role of TRPA1 triggering the migraine crisis in both sexes, whereas TRPM8 provided antinociceptive resilience exclusively in males. Notably, TRPM8 antinociception required the presence of testosterone and downregulation of this protective mechanism led to persistent hypersensitivity, whereas administration of testosterone to females favoured their recovery. TRPM8 also protected against formalin-induced pain, a model of acute pain dependent of TRPA1. Calcium imaging and electrophysiological recordings confirmed a testosterone activity on murine and human TRPM8 independent of androgen receptor expression.”

2)The state of art section could be improved.

The state-of-the-art section of the introduction was updated with several recent references and was modified to enhance comprehensibility as follows (page 3, paragraphs 1st-3rd):

“Chronic migraine is a highly prevalent and recurrent headache affliction particularly severe in women¹. While considerable advances have been made in the understanding of the disease (Ferrari et al., Nat Rev Dis Primers 2022; Iannone et al., Neurosci Lett 2022), the mechanisms underlying the sex dimorphism of chronic migraine remain largely unknown. A migraine model with high predictive validity is the sensory sensitization induced by nitroglycerin (NTG)⁴. Acute NTG treatments lead to a delayed hypersensitivity to mechanical stimulation that lasts hours in humans and rodents^{4,5}. Furthermore, repeated NTG exposure causes a chronic hypersensitivity that lasts several weeks in murine models⁶. This chronic hypersensitivity is characterized by generalized cutaneous sensitization, which has also been described as a reliable predictor of migraine chronification in humans⁷ (Han et al., Sci Rep 2021) that is found more prevalently in women (Bigal et al., Neurology 2008; Scher et al., Cephalalgia 2019).

Multiple migraine triggers including NTG have demonstrated a crucial involvement of transient receptor potential ankyrin 1 (TRPA1) in rodent models of acute migraine^{5,11}. In these acute models, NTG acts as a donor of nitric oxide triggering a TRPA1-mediated neuronal activity that promotes pain hypersensitivity (Marone et al., 2018). Indeed, TRPA1 is expressed in primary afferent neurons innervating the meninges where its activation favours the release of α -calcitonin gene-related peptide (α CGRP)^{11,12}, a neuropeptide that plays a pivotal role in migraine development^{4,13,14}. Thus, TRPA1 antagonists reduce meningeal inflammation and hypernociception in basic pain models (Iannone et al., Neurosci Lett 2022; Denner et al., Eur J Pain 2017) and α CGRP inhibition constitutes an effective strategy for migraine treatment in humans (Ferrari et al., Nat Rev Dis Primers 2022). Although recent reports have described higher contribution of α CGRP in females (Paige et al., J Neurosci 2022), the involvement of TRPA1 in males and females has not been yet studied in preclinical models of chronic migraine.

An additional TRP tightly associated with the expression of chronic migraine in humans is the transient receptor potential melastatin 8 (TRPM8). Several single-nucleotide polymorphisms affecting TRPM8 have been linked to chronic migraine and allodynia¹⁷ (Ling et al., J Headache Pain 2019). Concurrently, TRPM8 agonists such as menthol have been used medicinally for the alleviation of migraine-related^{19,20} and TRPA1-associated pain (Andersen et al., J Pain 2016).”

The following references were added:

- ✓ Andersen HH, Gazerani P, Arendt-Nielsen L. High-Concentration L-Menthol Exhibits Counter-Irritancy to Neurogenic Inflammation, Thermal and Mechanical Hyperalgesia Caused by Trans-cinnamaldehyde. *J Pain*. 2016 Aug;17(8):919-29. doi: 10.1016/j.jpain.2016.05.004. Epub 2016 May 31. PMID: 27260636.
- ✓ Bigal ME, Ashina S, Burstein R, Reed ML, Buse D, Serrano D, Lipton RB; AMPP Group. Prevalence and characteristics of allodynia in headache sufferers: a population study. *Neurology*. 2008 Apr 22;70(17):1525-33. doi: 10.1212/01.wnl.0000310645.31020.b1. PMID: 18427069; PMCID: PMC2664547.
- ✓ Ferrari MD, Goadsby PJ, Burstein R, Kurth T, Ayata C, Charles A, Ashina M, van den Maagdenberg AMJM, Dodick DW. Migraine. *Nat Rev Dis Primers*. 2022 Jan 13;8(1):2. doi: 10.1038/s41572-021-00328-4. PMID: 35027572.
- ✓ Han SM, Kim KM, Cho SJ, Yang KI, Kim D, Yun CH, Chu MK. Prevalence and characteristics of cutaneous allodynia in probable migraine. *Sci Rep*. 2021 Jan 28;11(1):2467. doi: 10.1038/s41598-021-82080-z. PMID: 33510340; PMCID: PMC7844001.
- ✓ Iannone LF, De Logu F, Geppetti P, De Cesaris F. The role of TRP ion channels in migraine and headache. *Neurosci Lett*. 2022 Jan 18;768:136380. doi: 10.1016/j.neulet.2021.136380. Epub 2021 Nov 30. PMID: 34861342.
- ✓ Ling YH, Chen SP, Fann CS, Wang SJ, Wang YF. TRPM8 genetic variant is associated with chronic migraine and allodynia. *J Headache Pain*. 2019 Dec 16;20(1):115. doi: 10.1186/s10194-019-1064-2. PMID: 31842742; PMCID: PMC6916225.
- ✓ Paige C, Plasencia-Fernandez I, Kume M, Papalampropoulou-Tsiridou M, Lorenzo LE, David ET, He L, Mejia GL, Driskill C, Ferrini F, Feldhaus AL, Garcia-Martinez LF, Akopian AN, De Koninck Y, Dussor G, Price TJ. A Female-Specific Role for Calcitonin Gene-Related Peptide (CGRP) in Rodent Pain Models. *J Neurosci*. 2022 Mar 9;42(10):1930-1944. doi: 10.1523/JNEUROSCI.1137-21.2022. Epub 2022 Jan 20. PMID: 35058371; PMCID: PMC8916765.
- ✓ Scher AI, Wang SJ, Katsarava Z, Buse DC, Fanning KM, Adams AM, Lipton RB. Epidemiology of migraine in men: Results from the Chronic Migraine Epidemiology and Outcomes (CaMEO) Study. *Cephalalgia*. 2019 Feb;39(2):296-305. doi: 10.1177/0333102418786266. Epub 2018 Jul 12. PMID: 29996667.

The aim of the study is not defined; this must be addressed.

The aim of the study was defined in page 4, lines 5-7:

“Here we implemented a murine model of chronic migraine that displays a sexual dimorphism characterized by enhanced pain sensitivity of females as described in humans (Ferrari et al., Nat Rev Dis Primers 2022). The aim of the study was to investigate the involvement of TRPA1 and TRPM8 in this sexually dimorphic behaviour. To address this, mechanical sensitivity was assessed...”

- ✓ Ferrari MD, Goadsby PJ, Burstein R, Kurth T, Ayata C, Charles A, Ashina M, van den Maagdenberg AMJM, Dodick DW. Migraine. *Nat Rev Dis Primers*. 2022 Jan 13;8(1):2. doi: 10.1038/s41572-021-00328-4. PMID: 35027572.

In light of this issue, to consider a possible revision of the title.

We considered a revision of the title stating the crucial involvement of TRPA1 in the model of chronic migraine. However, after finding previous reports describing the important role of TRPA1 in acute migraine (Marone et al., 2018; Benemei et al., 2014) we felt that the novelty of this result was not as strong as the pain-relieving effect of testosterone through TRPM8 and we finally decided to preserve the current title.

- ✓ Benemei S, Fusi C, Trevisan G, Geppetti P. The TRPA1 channel in migraine mechanism and treatment. *Br J Pharmacol*. 2014 May;171(10):2552-67. doi: 10.1111/bph.12512. PMID: 24206166; PMCID: PMC4008999.
- ✓ Marone IM, De Logu F, Nassini R, De Carvalho Goncalves M, Benemei S, Ferreira J, Jain P, Li Puma S, Bunnnett NW, Geppetti P, Materazzi S. TRPA1/NOX in the soma of trigeminal

ganglion neurons mediates migraine-related pain of glyceryl trinitrate in mice. *Brain*. 2018 Aug 1;141(8):2312-2328. doi: 10.1093/brain/awy177. PMID: 29985973; PMCID: PMC6061846.

3)The material and methods section is quite chaotic and difficult to follow. Bibliographic references of most of the dosage/timing/methods are missing (sections 2.2., 2.5., 2.6., 2.7., 2.8., 2.9., 2.12). Please provide where appropriate.

We made an effort to clarify and reorganize the methods section according to the order of presentation of the results. An Experimental design subsection containing the requested information was added in the main text (page 5, “2.3. Experimental design”) and was directed to Supplementary Methods.

The model of chronic migraine was moved ahead as subsection 2.2 (page 5); Subsection 2.4. “Assessment of mechanical sensitivity” (pages 5-6) was also re-organized; Information about tested drugs in behavioural and cellular studies was further clarified in subsections “2.8. Drugs tested in behavioural studies” (page 7) and “2.9. Drugs for cellular studies” (page 8).

References of dosage, timing and procedures were included in each subsection, and sentences were rephrased or further explained to increase clarity:

- Section **2.2** (now subsection 2.8, page 7):

“2.8. Drugs tested in behavioural studies.

*The TRPM8 selective blocker AMTB hydrochloride (AMTB, N-(3-aminopropyl)-2-[(3-methylphenyl) methyl] oxy}-N-(2-thienylmethyl) benzamide hydrochloride, Tocris, Bristol, UK) was dissolved in dimethyl sulfoxide (DMSO, Merck, Darmstadt, Germany) and was further diluted in saline to reach 2.5% DMSO. The range of AMTB doses (5-15 mg/kg i.p., 10 ml/kg) was chosen after dose conversion (Nair and Jacob, *J Basic Clin Pharm* 2016) from previous experiments demonstrating efficacy in attenuating hyperactive bladder activity in rats (Lashinger et al., 2008). Mechanical sensitivity was measured 30 min after AMTB, based in previous behavioural results in mice (Willis et al., *FASEB J* 2011).*

*The potent and selective TRPM8 agonist WS12 ((1R,2S,5R)-2-Isopropyl-N-(4-methoxyphenyl)-5-methylcyclohexanecarboxamide, Tocris) was dissolved in DMSO and diluted in corn oil to reach 2.5% DMSO. The range of WS12 i.p. doses (5-15 mg/kg i.p., 10 ml/kg) was chosen based on previous data showing antinociceptive effects of WS12 in mice (Liu et al., *Pain* 2013). WS12 was administered 30 min prior assessment of mechanical sensitivity (Liu et al., *Pain* 2013). In the formalin test, WS12 was dissolved in DMSO and diluted in saline up to 0.6% DMSO to achieve an amount of 6 nmol in 20 µl as previously described (Liu et al., *Pain* 2013). We used WS12 and not menthol or icilin as a TRPM8 agonist to avoid unspecific signalling over TRPA1⁴⁴.*

*Testosterone (T1500, Merck) was dissolved in 45% 2-Hydroxypropyl-β-cyclodextrin in water (Donner et al., *Endocrinology* 2016) to obtain a solution of 22 mg/ml for sustained delivery. This solution or its vehicle was introduced inside Alzet osmotic minipumps (Model 2004, Durect Corporation, Cupertino, CA, USA) to achieve a testosterone delivery of 6 µg/h, based on previous works obtaining significant effects in orchidectomized mice (Lesnak et al., *Pain* 2020; Zhao et al., *Nature* 2020). In the experiments assessing the acute effect of testosterone in females, testosterone was dissolved in DMSO and diluted in corn oil to reach 2.5% DMSO as with WS12, and it was administered at 1 mg/kg, 1 h before the assessment of mechanosensitivity based on previous data assessing its effects on anxiety-like behaviour (Frye et al., *Neuropsychopharmacology* 2008).”*

The new references were added to the reference list:

- ✓ Donner DG, Elliott GE, Beck BR, Bulmer AC, Lam AK, Headrick JP, Du Toit EF. Trenbolone Improves Cardiometabolic Risk Factors and Myocardial Tolerance to Ischemia-Reperfusion in

- Male Rats With Testosterone-Deficient Metabolic Syndrome. *Endocrinology*. 2016 Jan;157(1):368-81. doi: 10.1210/en.2015-1603. Epub 2015 Nov 19. PMID: 26584015.
- ✓ Lashinger ES, Steingina MS, Hieble JP, Leon LA, Gardner SD, Nagilla R, Davenport EA, Hoffman BE, Laping NJ, Su X. AMTB, a TRPM8 channel blocker: evidence in rats for activity in overactive bladder and painful bladder syndrome. *Am J Physiol Renal Physiol*. 2008 Sep;295(3):F803-10. doi: 10.1152/ajprenal.90269.2008. Epub 2008 Jun 18. PMID: 18562636.
 - ✓ Nair AB, Jacob S. A simple practice guide for dose conversion between animals and human. *J Basic Clin Pharm*. 2016 Mar;7(2):27-31. doi: 10.4103/0976-0105.177703. PMID: 27057123; PMCID: PMC4804402.
 - ✓ Willis DN, Liu B, Ha MA, Jordt SE, Morris JB. Menthol attenuates respiratory irritation responses to multiple cigarette smoke irritants. *FASEB J*. 2011 Dec;25(12):4434-44. doi: 10.1096/fj.11-188383. Epub 2011 Sep 8. PMID: 21903934; PMCID: PMC3236628.

- Section 2.5 (now subsection 2.6, page 6):

“2.6. Mouse Trigeminal Primary Cultures. *Adult mice (8-14 weeks) were sacrificed by cervical dislocation and trigeminal ganglia were extracted, micro-dissected from nerve projections and kept in ice-cold HBSS solution. Trigeminal ganglia were incubated in collagenase 48 µg/ml, 3.5 U/mg (C7657, Merck) and dispase 3 mg/ml, 1.79 U/mg (17105-041, Thermo Fisher Scientific, Waltham, USA) for 45 min at 37°C in 5% CO₂ (Ponsati et al., J Pharmacol Exp Ther. 2012). Thereafter, samples were mechanically disaggregated by pipetting 15 times through a 1 ml blue plastic tip. Neurons were separated from other cell types and tissue debris by placing the tissue homogenate over a 15% BSA solution (073k7601, Merck) and centrifuged 7 min at 0.3 RCF (Jocher et al., Front Cell Neurosci 2018). Then, cells were seeded in 5 µl drops over crystals treated with 8.3 µg/ml poly-L-Lysine (P9155, Merck) and 5 µg/ml laminin (L2020, Merck). 30 min after seeding cultures were supplemented with 500 µl of culture medium and incubated for 12-16 h at 37°C and 5% CO₂ (Ponsati et al., J Pharmacol Exp Ther. 2012). Cells were kept in a hormone-free culture medium consisting of Dulbecco's Modified Eagle Medium/F12 (DMEM/F12) without Phenol Red (11039021, Thermo Fisher Scientific), 1X Minimum Essential Medium (MEM) vitamin solution (11120052, Thermo Fisher Scientific), 1% penicillin/streptomycin (15140-022, Thermo Fisher Scientific) and home-made 4 µg/ml N₂-containing insulin (I2643, Merck), 0.1 mM putrescine hydrochloride (P7505, Merck), 3 nM sodium selenite (S5261 Merck), 100 µg/mL transferrin (T2872, Molecular Probes, Eugene, USA), 25 ng/ml NGF (G5141, Promega, Madison, USA) and 25 ng/ml hGDNF (450-10, Peprotech, London, UK).”*

These new references were added to the reference list:

- ✓ Jocher G, Mannschatz SH, Offterdinger M, Schweigreiter R. Microfluidics of Small-Population Neurons Allows for a Precise Quantification of the Peripheral Axonal Growth State. *Front Cell Neurosci*. 2018;12:166. Published 2018 Jun 15. doi:10.3389/fncel.2018.00166
- ✓ Ponsati B, Carreño C, Curto-Reyes V, et al. An inhibitor of neuronal exocytosis (DD04107) displays long-lasting in vivo activity against chronic inflammatory and neuropathic pain. *J Pharmacol Exp Ther*. 2012;341(3):634-645. doi:10.1124/jpet.111.190678

- Section 2.6 (now subsection 2.10, page 8):

“2.10. CGRP release assay. *Trigeminal cultures were incubated for 48h, then cells were treated with the exocytosis inhibitor 10 µM DD04107 (BCN Peptides SA, San Quintí de Mediona, Spain) or its vehicle (H₂O) for 1h (Ponsati et al., J Pharmacol Exp Ther. 2012). Next, cells were treated with 100 µM NTG or its vehicle (5% dextrose and 0.105% propylene glycol) for 30 additional min. Incubation solutions were made in culture medium and kept at 37°C and 5% CO₂.”*

- ✓ Ponsati B, Carreño C, Curto-Reyes V, et al. An inhibitor of neuronal exocytosis (DD04107) displays long-lasting in vivo activity against chronic inflammatory and neuropathic pain. *J Pharmacol Exp Ther.* 2012;341(3):634-645. doi:10.1124/jpet.111.190678

- Section 2.7 (now subsection 2.11, page 8):

“2.11. Immunocytochemistry. 30 min after NTG exposure (vehicle 5% dextrose and 0.105% propylene glycol), the media was removed from the cells and the culture was washed with 1X PBS (D8662, Merck) 3 times (Devesa et al., *Proc Natl Acad Sci U S A.* 2014). Afterwards, cells were fixed with 4% paraformaldehyde (158127, Merck) for 20 min at room temperature. Permeabilization was achieved with 0.1% v/v Triton 100X (P8787, Merck) for 5 min and blocking with 5% Normal Goat Serum (NGS, G9023, Merck) for 1 h, both in 1X PBS. Neurons were labelled with primary antibodies rabbit anti-MAP 1:250 (17490-1-AP, LabClinics, Barcelona, Spain) and mouse anti-CGRP 1:200 (AB81887, Abcam, Cambridge, UK) incubated overnight at 4°C. Secondary antibodies were goat anti-rabbit Alexa 488 1:1000 (A11034, Thermo Fisher Scientific) and goat anti-mouse Alexa 568 1:1000 (SAB4600400, Merck) were incubated for 1 h at room temperature and protected from light. Nuclei were stained with DAPI 1.5:10000 (D9564, Merck) for 5 min at room temperature. Slides were mounted with mowiol (475904, Merck) and images taken with a confocal microscope (LSM 900, ZEISS, Jena, Germany). Mean fluorescence intensity for each cell was obtained with ImageJ (Wayne Rasband, NIH), and the average value of positive cells was calculated for each picture.”

This new reference was added:

- ✓ Devesa I, Ferrándiz-Huertas C, Mathivanan S, et al. α CGRP is essential for algescic exocytotic mobilization of TRPV1 channels in peptidergic nociceptors. *Proc Natl Acad Sci U S A.* 2014;111(51):18345-18350. doi:10.1073/pnas.1420252111

- Section 2.8 (now subsection 2.12, page 8) was re-organized in the order of conducted experiments:

“2.12. Culture and transfection of human cell lines. Human embryonic kidney 293 cells (HEK293) were maintained in DMEM plus Glutamax, supplemented with 10% Fetal Bovine Serum (FBS, Thermo Fisher Scientific) and 1% penicillin/streptomycin, and incubated at 37°C in a 5% CO₂ atmosphere. For the study of TRPA1 function, HEK293 cells were plated in 24-well dishes at 2x10⁵ cells/well and transiently transfected with human TRPA1 in a pCMV6-AC-GFP vector (Viktorie Vlachova, Czech Academy of Sciences) using Lipofectamine 3000 (Thermo Fisher Scientific) (Nikolaeva-Koleva et al., *Sci. Rep.* 2021). For the transfection, 2 ml of Lipofectamine 3000 was mixed with the DNA in DMEM plus Glutamax with 1% FBS (Nikolaeva-Koleva et al., *Sci. Rep.* 2021). IMR90 fibroblast-like cells (CCL-186 ATCC, Virginia, USA) were seeded in 12 mm coverslips at 50000 cells/well and were maintained in MEM enriched with 10% FBS and penicillin/streptomycin 1% at 37°C in a 5% CO₂ atmosphere (Tonello et al., *Br J Pharmacol* 2017). For the initial experiments assessing TRPM8 function, HEK293 cells constitutively expressing human or rat TRPM8 (Journigan et al., *ACS Med Chem Lett* 202). were obtained from Prof. Belmonte Laboratory (Instituto de Neurociencias, San Juan, Alicante, Spain). All calcium imaging and electrophysiological recordings took place when cells reached 60% of confluence and 72 h after transfection when applicable.”

These references were added:

- ✓ Journigan VB, Alarcón-Alarcón D, Feng Z, Wang Y, Liang T, Dawley DC, Amin ARMR, Montano C, Van Horn WD, Xie XQ, Ferrer-Montiel A, Fernández-Carvajal A. Structural and in Vitro Functional Characterization of a Menthyl TRPM8 Antagonist Indicates Species-

Dependent Regulation. *ACS Med Chem Lett.* 2021 Mar 31;12(5):758-767. doi: 10.1021/acsmchemlett.1c00001. PMID: 34055223; PMCID: PMC8155240.

- ✓ Nikolaeva-Koleva M, Butron L, González-Rodríguez S, Devesa I, Valente P, Serafini M, Genazzani AA, Pirali T, Ballester GF, Fernández-Carvajal A, Ferrer-Montiel A. A capsaicinoid-based soft drug, AG1529, for attenuating TRPV1-mediated histaminergic and inflammatory sensory neuron excitability. *Sci Rep.* 2021 Jan 8;11(1):246. doi: 10.1038/s41598-020-80725-z. PMID: 33420359; PMCID: PMC7794549.
- ✓ Tonello R, Fusi C, Materazzi S, Marone IM, De Logu F, Benemei S, Gonçalves MC, Coppi E, Castro-Junior CJ, Gomez MV, Geppetti P, Ferreira J, Nassini R. The peptide Phc1 β , from spider venom, acts as a TRPA1 channel antagonist with antinociceptive effects in mice. *Br J Pharmacol.* 2017 Jan;174(1):57-69. doi: 10.1111/bph.13652. Epub 2016 Nov 28. PMID: 27759880; PMCID: PMC5341489.

- Section 2.9 (now subsection 2.7, page 7):

“2.7. Fluorescence Ca²⁺ imaging.”

“Trigeminal neurons or Human Embryonic Kidney 293 (HEK293) cells (CRL-1573 ATCC) were incubated with 5 mM (6 mg/ml) fluo4-AM and 0.2% w/v pluronic acid (F-127, Thermo Fisher Scientific) for 60 min at 37°C in standard extracellular solution (in mM: 140 NaCl, 3 KCl, 2.4 CaCl₂, 1.3 MgCl₂, 10 HEPES, and 5 glucose, adjusted to pH 7.4 with 1M NaOH) (Devesa et al., *Proc Natl Acad Sci U S A.* 2014).”

“Before starting the experiment, an image of the microscopic field was obtained with transmitted light to create the regions of interest around cells with neuronal morphology.”

“Responses were scored as positive if the increase in fluorescence was >0.2 arbitrary units (Devesa et al., *Proc Natl Acad Sci U S A.* 2014).”

- ✓ Devesa I, Ferrándiz-Huertas C, Mathivanan S, et al. α CGRP is essential for algescic exocytotic mobilization of TRPV1 channels in peptidergic nociceptors. *Proc Natl Acad Sci U S A.* 2014;111(51):18345-18350. doi:10.1073/pnas.1420252111

- Section 2.12 (now subsection 2.14):

“2.14. **Orchidectomy.** Orchidectomy was adapted from a previous procedure (Sophocleous et al., *Methods Mol Biol* 2019) changing to a single midline scrotal incision to minimize tissue injury. Briefly, mice were anesthetized with a mixture of i.p. ketamine (75 mg/kg; Imalgene, 100 mg/ml, Boehringer Ingelheim, Ingelheim/Rhein, Germany) and xylazine (15 mg/kg, Merck) and a single midline scrotal incision was made.”

New reference:

- ✓ Sophocleous A, Idris AI. Ovariectomy/Orchiectomy in Rodents. *Methods Mol Biol.* 2019;1914:261-267. doi:10.1007/978-1-4939-8997-3_13

4) Please revise the material and methods section to better describe the experimental groups/sets and the specific number of animals used for each evaluation (NTG model and mechanical sensitivity, trigeminal cultured cells to CGRP assay and IHC, ect). The experimental design is not clear: a brief description of the timing of each assessment would be useful to better follow all experimental evaluations; tables and/or schemes may be of help.

This information was included in the new subsection 2.3. **Experimental Design** (Page 5, 3rd paragraph), which is re-directed to the Supplementary Methods, where the required information was included:

“2.3. Experimental design. Detailed description of the rationale, timing and experimental design followed in each experiment can be found in Supplementary Methods.”

Tables/schemes have also been added to the Supplementary Methods to facilitate comprehension. Information on number of animals and experimental groups can also be found in the Source Data File for the behavioural experiments.

5) Section 2.2: to explain why two formulations of NTG were used.

Explanation was included in subsection 2.2 (page 5, 2nd paragraph):

“Two different NTG formulations were tested: first, 5 mg/1.5 ml ampoules and second, 50 mg/50 ml vials (Bioindustria LIM, Novi Ligure, Italy). The 5 mg/1.5 ml ampoules contained NTG dissolved on a vehicle made of 1 ml propylene glycol and 0.5 ml ethanol (Bioindustria LIM). Such solution was further dissolved in saline to obtain 1 mg/ml NTG, reaching final concentrations of 10% ethanol and 20% propylene glycol. This preparation was no longer used after the first behavioural experiment because the treatment with vehicle induced mechanical pain sensitization by itself as shown in Supplementary Figure 1. The second NTG formulation consisted of 50 mg/50 ml NTG vials containing a vehicle made of 5% dextrose and 0.105% propylene glycol in pure water (Bioindustria LIM). This latter NTG formulation or its vehicle was administered without further dilution. Such preparation was chosen to complete all the remaining experiments because the vehicle did not modify nociceptive sensitivity. Both the first and the second preparations were administered intraperitoneally (i.p.) at 10 ml/kg, obtaining the same NTG dose of 10 mg/kg as previously described (Pradhan et al., Pain 2014).”

6) Why did the Authors opt for the evaluation of mechanical allodynia in the hind paw and not in the trigeminal area?

Evaluation in the trigeminal area was considered, however it was disregarded because of several limitations. First, evaluation of mechanical allodynia in the trigeminal region requires a manipulation of the animal that includes shaving of the periorbital area and restraint of the mice as previously described (Elliott et al., *Headache* 2012). Since males and females have different sizes, different restraining devices should be used. In addition, immobilization represents a source of stress and can constitute by itself a stress model that could have significant impact on the progress of nociceptive behaviour (Deslauriers et al., *Biol Psychiatry* 2018; Avona et al., *Pain* 2020; Ji et al., *J Pain* 2018). Second, nociceptive evaluation of the mice requires application of the filaments when the animals are not attending and this may be rather difficult when the application is near the eyes of the animal. By application in the paw, the influence of hypervigilance/affective status on subsequent nociceptive evaluations may be mitigated.

Furthermore, other models of migraine based on the selective activation of trigeminal afferents also trigger generalized mechanical hypersensitivity (Avona et al., *J Headache Pain* 2021). Since the chronic and systemic nitroglycerin administrations elicited such generalized mechanical sensitivity in a sexual dimorphic fashion, we decided to focus on that behaviour as a proxy of the sexual dimorphism also described during chronic migraine in humans (Bigal et al., 2008).

- ✓ Avona A, Mason BN, Lackovic J, Wajahat N, Motina M, Quigley L, Burgos-Vega C, Moldovan Loomis C, Garcia-Martinez LF, Akopian AN, Price TJ, Dussor G. Repetitive stress in mice causes migraine-like behaviors and calcitonin gene-related peptide-dependent hyperalgesic priming to a migraine trigger. *Pain*. 2020 Nov;161(11):2539-2550. doi: 10.1097/j.pain.0000000000001953. PMID: 32541386; PMCID: PMC7572536.
- ✓ Avona A, Price TJ, Dussor G. Interleukin-6 induces spatially dependent whole-body hypersensitivity in rats: implications for extracephalic hypersensitivity in migraine. *J Headache*

- Pain. 2021 Jul 13;22(1):70. doi: 10.1186/s10194-021-01286-8. PMID: 34256692; PMCID: PMC8278737.
- ✓ Bigal ME, Ashina S, Burstein R, Reed ML, Buse D, Serrano D, Lipton RB; AMPP Group. Prevalence and characteristics of allodynia in headache sufferers: a population study. *Neurology*. 2008 Apr 22;70(17):1525-33. doi: 10.1212/01.wnl.0000310645.31020.b1. PMID: 18427069; PMCID: PMC2664547.
 - ✓ Deslauriers J, Toth M, Der-Avakian A, Risbrough VB. Current Status of Animal Models of Posttraumatic Stress Disorder: Behavioral and Biological Phenotypes, and Future Challenges in Improving Translation. *Biol Psychiatry*. 2018 May 15;83(10):895-907. doi: 10.1016/j.biopsych.2017.11.019. Epub 2017 Nov 20. PMID: 29338843; PMCID: PMC6085893.
 - ✓ Elliott MB, Oshinsky ML, Amenta PS, Awe OO, Jallo JI. Nociceptive neuropeptide increases and periorbital allodynia in a model of traumatic brain injury. *Headache*. 2012 Jun;52(6):966-84. doi: 10.1111/j.1526-4610.2012.02160.x. Epub 2012 May 8. PMID: 22568499; PMCID: PMC4105160.
 - ✓ Ji Y, Hu B, Li J, Traub RJ. Opposing Roles of Estradiol and Testosterone on Stress-Induced Visceral Hypersensitivity in Rats. *J Pain*. 2018 Jul;19(7):764-776. doi: 10.1016/j.jpain.2018.02.007. Epub 2018 Mar 2. PMID: 29496640; PMCID: PMC6026052.

7) Why did the Authors perform plantar formalin test instead of the orofacial version?

Since the von Frey test conducted in the hind paw yielded nonsignificant antinociception after WS12, our first objective with the formalin test in the paw was to distinguish whether WS12 could elicit antinociceptive effects at all in the same body area after a TRPA1-dependent pain. Similarly, we tested AMTB in male mice recovered from the formalin test in the paw as a way to provide further evidence of the endogenous TRPM8 activity that provided inhibition of paw hypersensitivity in mice previously exposed to NTG.

And more importantly, why didn't they combine the formalin test with the NTG model?

The ethical committee of our institution does not allow the combination of two models purposely conducted to induce pain or suffering to the animals. When animals need to be exposed to two procedures deliberately made to induce pain, they must be allowed to recover from the first insult before the following one can be applied.

8)Section 2.11: to include timing and dosage of TRPM8 agonist and antagonist. To motivate why the TRPM8 agonist was given before and the agonist after formalin injection.

This information was included in subsection 2.13 (before 2.11, page 9) as follows:

“2.13. Chemically-induced nocifensive behaviour (Formalin test). Mice were individually placed into transparent chambers and were habituated for 1 h before testing. To evaluate the pain-relieving effect of WS12 against formalin-induced nocifensive behaviour, 20 µl of a 45% 2-Hydroxypropyl-β-cyclodextrin solution containing 5% formalin (F8775, Merck) and 0.6% DMSO with or without 6 nmol of WS12 were injected subcutaneously into the plantar aspect of the right hind paw (i.pl.) by using a Hamilton syringe (Hamilton Syringe Gastight™ serie 1700, TLL end, Merck) coupled to a 30-gauge needle. The time spent expressing nocifensive behaviour (licking or biting of injected paw) was quantified in 5 min intervals during 60 min as previously described⁶⁵. To assess the influence of endogenous TRPM8 activity on the extinction of nocifensive behaviour after formalin, mice received first 20 µl of 5% formalin dissolved in saline in their right hind paw (i.pl.), and nocifensive behaviour was quantified as above. After the initial 60 min when the nocifensive responses ceased, AMTB (10 mg/kg) or vehicle (2.5% DMSO in saline) were administered i.p. and behavioural quantification continued for 1 h.”

Rationale, timing and dosages are included in Supplementary Methods, pages 5 and 6, S1.8 and S1.10.:

“S1.8. Evaluation of the pain-relieving efficacy of TRPM8 agonism in formalin-sensitized female mice. To rule out possible antinociceptive effects of WS12 in females subjected to TRPA1-mediated pain, the effect of WS12 was evaluated after local administration (i.pl.) in the formalin test, a TRPA1-dependent model of acute pain. Formalin was co-injected with 6 nmol of WS12 (n=4) or its vehicle (n=7) in the right hind paw of wild-type females, and licking behaviour was quantified for 1 h (Figure 3F).”

Experimental Groups			
Sex	Genotype	Treatment (i.pl.)*	Mice tested
Female	WT	5% Formalin in vehicle	7
		5% Formalin in 6 nmol WS12 in vehicle	4

* Vehicle: 0.6% DMSO in 45% 2-Hydroxypropyl- β -cyclodextrin in water

Time (min)	Time line behavioural experiment													
	0	5	10	15	20	25	30	35	40	45	50	55	60	
Treatment 1														
Quantification of licking behaviour														

“S1.10. Evaluation of endogenous protective TRPM8 activity in formalin-exposed male mice. An additional experiment was conducted to investigate the presence of endogenous protective TRPM8 tone after formalin-induced pain in males (Figure 3H). Males were exposed to 5% i.pl. formalin and licking behaviour was quantified for 1 h until cessation of nocifensive behaviour. Then, mice received AMTB 10 mg/kg i.p. (n=6) or its vehicle (n=6) and quantification of licking behaviour continued for 1 h.”

Experimental Groups				
Sex	Genotype	Treatment 1 (i.pl.)* ¹	Treatment 2 (i.p.)* ²	Mice tested
Male	WT	5% Formalin	Vehicle	6
			AMTB 10 mg/kg	6

*¹ Vehicle: Saline

*² Vehicle: 2.5% DMSO in saline

Time (min)	Time line behavioural experiment																									
	0	5	10	15	20	25	30	35	40	45	50	55	60	65	70	75	80	85	90	95	100	105	110	115	120	
Treatment 1																										
Quantification of licking behaviour																										
Treatment 2																										

9)Section 2.12:to provide a description of drugs treatment (NTG, TRPM8 antagonist, etc) of the orchidectomized mice.

This information is now included in the Supplementary Methods, Experimental design, pages 6-7, **S1.12. Elucidation of testosterone and TRPM8 involvement on the recovery of normal sensitivity observed in male mice after cessation of the chronic NTG treatment:**

“...all mice were orchidectomized. 3 weeks later, mechanical sensitivity was again assessed and mice were implanted with alzet osmotic minipumps (Model 2004, Durect Corporation, Cupertino, CA, USA) filled with testosterone or its vehicle. 4 experimental groups with orchidectomized males were obtained: wild-type mice receiving vehicle (n=5) or testosterone infusions (n=6, 6 μ g/h) and TRPM8 KO mice receiving vehicle (n=6) or testosterone (n=7). 3 days later, mechanical sensitivity was assessed and males were exposed to the chronic treatment with NTG 10 mg/kg or its vehicle (i.p., days 0, 2, 4, 6, 8). Mechanical sensitivity was measured (before and 2 h after NTG injections on days 0 and 8, and on days 14 and 20). To further discriminate the involvement of TRPM8 on the pain-relieving effects of exogenous testosterone, all mice were exposed to a dose

of AMTB (10 mg/kg i.p.) on day 20 and mechanical sensitivity was evaluated again 30 min later (Figure 4C).

Experimental Groups						
Sex	Genotype	Surgery	Treatment 1 (Minipump)* ¹	Treatment 2 (i.p.)* ²	Treatment 3 (i.p.)* ³	Mice tested
Male	WT	Orchidectomy	Vehicle	NTG 10 mg/kg	AMTB 10 mg/kg	5
			Testosterone 6 µg/h			6
	TRPM8KO		Vehicle			6
			Testosterone 6 µg/h			7

*¹ Vehicle: 45% 2-Hydroxypropyl-β-cyclodextrin in water

*² Vehicle: 5% dextrose, 0.105% propylene glycol in water

*³ Vehicle: 2.5% DMSO in saline

Time (day)	Time line behavioural experiment																									
	-17	-16	-15	-14	...	-4	-3	...	0	1	2	3	4	5	6	7	8	9	...	13	14	15	...	19	20	
Habituation																										
Surgery																										
Minipump implantation																										
Von Frey (No treatment on this day)																										
Treatment 1																										
Von Frey right before treatment																										
Treatment 2																										
Von Frey 2h after treatment 2																										
Treatment 3																										
Von Frey 30 min after treatment 3																										

”

10) At what cycle phase were female mice evaluated for behavioral outputs? Did the authors use vaginal cytology to verify the estrous cycle stage?

We used vaginal cytology to evaluate the possible influence of estrous cycle in the nociceptive sensitization induced after NTG. To do that, female mice treated with NTG 10 mg/kg or vehicle were evaluated for mechanical sensitivity on day 14 and vaginal cytology was conducted as previously described (McLean et al., 2012). Data from NTG and vehicle-treated mice were subdivided into that of mice in stages of low estrogen levels (estrus) and that of mice in stages with high estrogen levels (proestrus, metestrus and diestrus), as previously described (Escudero-Lara et al., 2021; Nilsson et al., 2015; Zenclussen et al., 2014). Since no trend of a significant effect of estrous cycle was observed (See graph below, P=0.1305 for cycle effect) we concluded that cycle stage was at least not a principal determining factor of hypersensitivity in these experimental conditions, and the use of vaginal cytology ceased to prevent possible interferences of vaginal stimulation on animal behaviour.

Day 14 (6 days after chronic treatment)

N (High estrogen Vehicle: 8; Low estrogen Vehicle: 3; High estrogen NTG: 7; Low estrogen NTG: 3)

- ✓ Escudero-Lara A, Cabañero D, Maldonado R. Kappa opioid receptor modulation of endometriosis pain in mice. *Neuropharmacology*. 2021 Sep 1;195:108677. doi: 10.1016/j.neuropharm.2021.108677. Epub 2021 Jun 19. PMID: 34153313.
- ✓ McLean AC, Valenzuela N, Fai S, Bennett SA. Performing vaginal lavage, crystal violet staining, and vaginal cytological evaluation for mouse estrous cycle staging identification. *J Vis Exp*. 2012 Sep 15;(67):e4389. doi: 10.3791/4389. PMID: 23007862; PMCID: PMC3490233.
- ✓ Nilsson ME, Vandenput L, Tivesten Å, Norlén AK, Lagerquist MK, Windahl SH, Börjesson AE, Farman HH, Poutanen M, Benrick A, Maliqueo M, Stener-Victorin E, Ryberg H, Ohlsson C. Measurement of a Comprehensive Sex Steroid Profile in Rodent Serum by High-Sensitive Gas Chromatography-Tandem Mass Spectrometry. *Endocrinology*. 2015 Jul;156(7):2492-502. doi: 10.1210/en.2014-1890. Epub 2015 Apr 9. PMID: 25856427.
- ✓ Zenclussen ML, Casalis PA, Jensen F, Woidacki K, Zenclussen AC. Hormonal Fluctuations during the Estrous Cycle Modulate Heme Oxygenase-1 Expression in the Uterus. *Front Endocrinol (Lausanne)*. 2014 Mar 13;5:32. doi: 10.3389/fendo.2014.00032. PMID: 24659985; PMCID: PMC3952397.

11)Section 2.15: the sample size calculation and distribution data should be included.

(Section 2.15 is now 2.20) Sample size estimation was added in the Methods section, page 12, last lines of Subsection 2.20. Statistical Analyses:

“*Sample size for animal and cellular studies was estimated based on previous experience in our laboratories using similar behavioural and cellular approaches (Cabañero et al., Elife 2020; Devesa et al., Proc Natl Acad Sci U S A. 2014, Nikolaeva et al., Sci Rep. 2021).*”

- ✓ Nikolaeva-Koleva M, Butron L, González-Rodríguez S, Devesa I, Valente P, Serafini M, Genazzani AA, Pirali T, Ballester GF, Fernández-Carvajal A, Ferrer-Montiel A. A capsaicinoid-based soft drug, AG1529, for attenuating TRPV1-mediated histaminergic and inflammatory sensory neuron excitability. *Sci Rep*. 2021 Jan 8;11(1):246. doi: 10.1038/s41598-020-80725-z. PMID: 33420359; PMCID: PMC7794549.
- ✓ Cabañero D, Ramírez-López A, Drews E, Schmöle A, Otte DM, Wawrzczak-Bargiela A, Huerga Encabo H, Kummer S, Ferrer-Montiel A, Przewlocki R, Zimmer A, Maldonado R. Protective role of neuronal and lymphoid cannabinoid CB₂ receptors in neuropathic pain. *Elife*. 2020 Jul 20;9:e55582. doi: 10.7554/eLife.55582. PMID: 32687056; PMCID: PMC7384863.

- ✓ Devesa I, Ferrándiz-Huertas C, Mathivanan S, et al. α CGRP is essential for algesic exocytotic mobilization of TRPV1 channels in peptidergic nociceptors. *Proc Natl Acad Sci U S A.* 2014;111(51):18345-18350. doi:10.1073/pnas.1420252111.

Distribution data and additional missing information related to statistical analyses were included at the end of subsection 2.20 as follows:

“Throughout the manuscript, data are expressed as mean \pm SEM values and individual data points are shown whenever possible. Differences were considered statistically significant when P values were less than 0.05. Statistical analyses were performed using IBM SPSS 25 Software (IBM, Chicago, IL, USA) and GraphPad Prism 7.4 (GraphPad Software Inc., San Diego, CA, USA). Additional distribution data can be found in the Supplementary Statistical Analysis Data File.”

Distribution data can also be found in the Suppl. Material Statistical data file and in the Source Data File for the behavioural studies.

12)The timing of Figure 1 is confusing. Please use consistency for days’ numbers (panel A vs plots and figure legend).

Timing of Figure 1 was amended:

13)All figures showing NTG-induced allodynia in males and females’ mice (acute and chronic NTG) should be displayed in a single graph.

Figures showing NTG-induced allodynia in males and females are now displayed in single graphs as requested.

Figure 1B:

Supplementary Figure 1:

Figure 2A:

Figure 3A:

14)The organization of results are quite confusing, it is lacking a logical sequence of the data presented. Moreover, it is not clear why in some circumstances the same evaluations were performed only in females or males. For instance, AMTB within the formalin test was only tested in male mice.

The results have been reorganized and the rationale for each experiment is now specified in the Experimental Design subsection (subsection 2.3 of Methods, page 5, directed to supplementary material).

In addition, in results subsection 3.1. **Repeated NTG treatment induces a persistent hypersensitivity exclusively expressed in female mice** (page 13), we are starting the results with the description of the administration protocol (page 13, beginning of 1st paragraph, 2.1 subsection of Results section)

“NTG (10 mg/kg) or vehicle (5% dextrose and 0.105% propylene glycol in water) were administered intraperitoneally (i.p.) every other day during 8 days to male and female mice, and mechanical sensitivity was assessed before and after each treatment and for 12 additional days after the end of the repeated administrations (Figure 1A).”

and the results with the effects of the vehicle that produces mechanical hypersensitivity have been left in the methods section (page 5, subsection **2.2. Model of chronic migraine**):

“Two different NTG formulations were tested: first, 5 mg/1.5 ml ampoules and second, 50 mg/50 ml vials (Bioindustria LIM, Novi Ligure, Italy). The 5 mg/1.5 ml ampoules contained NTG dissolved on a vehicle made of 1 ml propylene glycol and 0.5 ml ethanol (Bioindustria LIM). Such solution was further dissolved in saline to obtain 1 mg/ml NTG, reaching final concentrations of 10% ethanol and 20% propylene glycol. This preparation was no longer used after the first behavioural experiment because the treatment with vehicle induced mechanical pain sensitization by itself as shown in Supplementary Figure 1.”

By this way, the results paragraph is now focused on the main result obtained with the nitroglycerin with innocuous vehicle.

The assessment of AMTB effect in males recovered from formalin-induced pain was conducted to corroborate the existence of a TRPM8-dependent tone providing relief after TRPA1-induced pain in males, as previously observed in the NTG model. Since females remained sensitized in the NTG model and lacked that protection, those experiments were only conducted in males to optimize animal use. This explanation was included in page 15, at the end of the last paragraph of Results subsection **“3.3. TRPM8 activity determines the recovery of normal sensitivity in male mice exposed to the model of chronic migraine”**:

“To corroborate the existence of a TRPM8-dependent tone providing relief after TRPA1-induced pain in males, male mice were treated with AMTB after the extinction of formalin-induced nocifensive activity. These mice showed...”

15)The discussion section is also a bit confused. It is not clear whether the involvement of TRPM8 is related to its activation or desensitization.

This has been clarified at the beginning of the Discussion section as follows (page 19, 1st paragraph):

“The most salient contribution of our study is the discovery of the role of TRPM8 as an androgen-activated receptor that provides antinociceptive resilience in a sex-dependent manner. Specifically, TRPM8 activity favours recovery in a mouse model of NTG-induced chronic migraine that produces similar acute mechanosensitivity in males and females but persistent chronic hypersensitivity exclusively in females. Our results suggest that testosterone by activating TRPM8 channels drives a sexual dimorphism characterized by recovery of normal sensitivity in males.”

Our results suggest a pain relief mediated through TRPM8 activation. On the one hand TRPM8 deletion worsens chronic sensitization in NTG-exposed males, and if desensitization of the channel would drive pain relief one would expect to observe antinociceptive effect in the knockout. In the same line, TRPM8 antagonism with AMTB produces re-sensitization in both formalin and NTG models, suggesting an endogenous TRPM8 activity maintaining normal sensitivity after these insults.

By using a TRPM8 agonist the Authors found no effect in female mice, thus it is unclear in which way this channel may operate in the prolonged chronic sensitization induced by NTG.

The TRPM8 agonist WS12 did not induce significant relief of mechanical pain in the NTG model. We believe that this could be due to the intrinsic characteristics of the nociceptive evaluation and/or to the pharmacokinetics/pharmacodynamics of the compound.

On the one hand, as stated, the application of von Frey filaments in freely behaving mice must be spaced to avoid hypervigilance or affectation of the nociceptive responses by previous stimulations. Since the animals also need to be in an adequate behavioural state in order to applying the stimuli and obtaining a reliable response (Callahan et al., *J Pain* 2008), a range of 15-30 min is the minimum necessary to apply the complete filament sequence and to obtain consistent behavioural responses in subgroups of 6-7 mice. Hence, if WS12 effects last ~5 min, the response to some filaments will be sensitive to the antinociceptive effect, but there may not be enough time to apply the complete sequence of filaments under the effect of the drug. Although, the administration of testosterone did induce a significant antinociceptive effect that was disrupted in TRPM8 knockout females, suggesting a pain-relieving effect of TRPM8 activity that depends on the type of activation or the pharmacokinetics of the drug.

- ✓ Callahan BL, Gil AS, Levesque A, Mogil JS. Modulation of mechanical and thermal nociceptive sensitivity in the laboratory mouse by behavioral state. *J Pain*. 2008;9(2):174-184. doi:10.1016/j.jpain.2007.10.011

As appreciated in the formalin test, a significant WS12 effect can be found in the first 5 min after its local injection in the site of action. However, the intraperitoneal administration of WS12 in the NTG model requires systemic distribution of the drug before obtaining an antinociceptive effect. While testosterone can be distributed through endogenous carriers such as albumin or the sex hormone binding globulin (SHBG), WS12 lacks of known plasmatic transporters and it is highly lipophilic and poorly hydrosoluble, qualities that could result in poor distribution and low efficiency. These aspects have been highlighted in the Discussion section as follows (page 20, end of 2nd paragraph):

“In this line, short-lasting pain-relieving responses to menthol and its derivatives can be found clinically and were described elsewhere⁶⁶. On the other hand, the poor hydrosolubility of WS12 required its dissolution in corn oil for systemic administration and the lack of a significant effect could also be related to an erratic absorption and distribution of the compound.”

Regarding the way TRPM8 could provide antinociceptive resilience, possible mechanisms have been included in Discussion section, end of page 19- beginning of page 20:

*“Several mechanisms could underlie TRPM8-mediated antinociception. On the one hand, it has been proposed that TRPM8 activation produces silencing of C nociceptors expressing TRPA1 (Liu et al., *J Appl Physiol* 2015) through interneuron recruitment in the spinal cord dorsal horn (Kayama et al., *Cephalalgia* 2018; Ren, Dhaka, and Cao, *Mol. Pain* 2015) or via inhibitory metabotropic glutamatergic receptors expressed in the central terminal of nociceptors (Proudfoot et al., *Curr Biol* 2006). Thus, in the studied pain models TRPM8 activity may occlude signalling of TRPA1-expressing nociceptors and subsequent CGRP release (Liu et al., *J Appl Physiol* 2015). In addition, a participation of central brain areas cannot be excluded from our data, given the recent description of TRPM8 also in supraspinal locations (Ordás, *J Comp Neurol* 2021).”*

The new references were added:

- ✓ Liu BY, Lin YJ, Lee HF, Ho CY, Ruan T, Kou YR. Menthol suppresses laryngeal C-fiber hypersensitivity to cigarette smoke in a rat model of gastroesophageal reflux disease: the role of TRPM8. *J Appl Physiol* (1985). 2015;118(5):635-645. doi:10.1152/jappphysiol.00717.2014
- ✓ Proudfoot CJ, Garry EM, Cottrell DF, et al. Analgesia mediated by the TRPM8 cold receptor in chronic neuropathic pain. *Curr Biol*. 2006;16(16):1591-1605. doi:10.1016/j.cub.2006.07.061.

Reviewer #2 (Remarks to the Author):

1.- As a matter of style, I suggest not starting the Results with the effects of the vehicle that produces mechanical hypersensitivity. This is interesting, but it is just a side detail. I would propose to just make a small comment about this. In fact, the effects of the vehicles should be presented after describing the administration protocol (We injected 10 mg/kg NTG or vehicle...).

As suggested, we are now starting the Results with the description of the NTG administration protocol (page 13, beginning of 1st paragraph, 3.1 subsection of Results section)

“NTG 10 mg/kg or vehicle (5% dextrose and 0.105% propylene glycol in water) were administered intraperitoneally (i.p.) every other day during 8 days to male and female mice, and mechanical sensitivity was assessed before and after each treatment and for 12 additional days after the end of the repeated administrations (Figure 1A).”

And the comment of the vehicle producing mechanical hypersensitivity was left in methods section (page 5, 2nd paragraph, subsection 2.2. Model of chronic migraine):

“The 5 mg/1.5 ml ampoules contained NTG dissolved on a vehicle made of 1 ml propylene glycol and 0.5 ml ethanol (Bioindustria LIM). Such solution was further dissolved in saline to obtain 1 mg/ml NTG, reaching final concentrations of 10% ethanol and 20% propylene glycol. This preparation was no longer used after the first behavioural experiment because the treatment with vehicle induced mechanical pain sensitization by itself as shown in Supplementary Figure 1.”

2.- Line 405, the term “calcium imaging activity” is not correct.

The term was changed to *“intracellular calcium imaging”* (page 13, beginning of last paragraph).

3.- In my opinion, the results of the intracellular Ca²⁺ imaging experiments are difficult to interpret. Ideally, direct (ratiometric) measurements of intracellular Ca²⁺ should have been performed to be able to compare the amplitude of signals elicited in neurons from different animals. Differences may be attributed to distinct loading of the dye.

It is unclear whether the normalization to the amplitude of the responses to high K⁺ is appropriate. For this to work, one has to assume that the treatments do not alter the expression and function of voltage-gated Ca²⁺ channels, the resting potential or several other factors that determine the response to high K⁺.

We agree with the reviewer that ratiometric probes would allow a comparison between experiments of higher consistency. Unfortunately, our fluorescence setup does not allow for rapid switching between two excitatory wavelengths, hence we need to use an external positive control such as potassium chloride to normalize the data, with the assumption that this response is not altered by the treatment of study.

To rule out possible differences in potassium chloride responses underlying the differences in the transients elicited by AITC, we quantified the raw fluorescence change produced by 40mM KCl (**see figure below**) in all the neurons (**panel A**) and in AITC-responsive neurons (**panel B**) from mice treated with vehicle or NTG. We found similar distribution with no statistical differences in all neurons (**Figure a**, nonsignificant differences with t-test, n=5 mice per group, 1198 cells in vehicle group, 1211 cells NTG group) or AITC-responsive neurons (**Figure b**, nonsignificant differences with t-test, n=5 mice per group, 246 cells vehicle group, 366 cells NTG group).

We added a sentence in the Results section to acknowledge that the results presented are obtained with non-ratiometric calcium imaging (3.2 subsection, page 13, last paragraph):

“These results obtained with non-ratiometric calcium imaging were obtained by normalization of AITC-induced transients to 40 mM KCl responses which were similar in vehicle and NTG-treated mice.”

It is unclear how the statistical analysis on Ca²⁺ imaging was performed. What are exactly the values presented in Fig. 2E, 2G, 3E and 4F?

The data sets of each calcium imaging experiment were analysed following this rationale: First, a D’Agostino-Pearson normality test is applied. When the data follows a normal distribution, data are analysed with t-tests (2 groups) or ANOVA tests followed by Bonferroni post-hoc comparisons (more than 2 groups). When the data is found to be not normal, Mann-Whitney tests (2 groups) or Kruskal Wallis (more than 2 groups) followed by Mann-Whitney tests are used. This is explained in Methods, Subsection 2.20. Statistical Analyses (page 12, lines 15-21 of the 2nd paragraph).

In the calcium imaging experiments of primary cultures, 2 parameters are studied: the percentage of responsive neurons relative to the total number of neurons, which is estimated by the response to KCl, and the amplitude of the calcium transient elicited in responsive neurons normalized to the KCl response. In the graphs showing the size of the calcium transients, each data point of the graphs represents the average of the size of the transients obtained in one animal. The number of cells measured per animal ranges from 100 to 200 neurons.

In figures 2E (now 2D) and 3E (now 3D) the data show average response sizes normalized to KCl. The response size to AITC or WS12 is calculated by subtracting the basal values of fluorescence to the peak reached after stimulus application, and then normalizing to the KCl response. The response sizes of all the responsive neurons from one animal are averaged. Finally, the mean data obtained from each animal is used for the statistical analysis between experimental groups. In figures 2E and 3E, mean values for 5 male and 5 female mice treated with vehicle or NTG are compared by 2-way ANOVA with sex and treatment as between-subject factors.

We added the following phrases to the legends to clarify what are exactly the values in these figures (Legends of figures 2D and 3D):

2D *“AITC response size of trigeminal cultured neurons, each data point represents the average response size of NTG-sensitive neurons vs their respective KCl responses, in one animal.”*

3D “*WS12 response size of trigeminal cultured neurons, each data point represents the average response size of WS12-sensitive neurons vs their respective KCl responses, in one animal.*” A title was added to this figure panel: “*WS12 response size*”.

In figures 2G (now 2F) and 4E data values represent percentage of neurons responding to the presented stimuli against the total KCl-responding neurons. In figures 2F and 4E, we quantify the percentage of neurons that respond to both stimuli (2F AITC&NTG, 4E Testosterone&WS12). Here we compare 4 wild type and 4 knockout mice, and the percentage of cells responding vs the cells responding to KCl in each animal is represented as a data point. Since the data does not follow a normal distribution, the results of the % of responses to the stimuli are compared by Mann-Whitney U.

We added the following phrases in the legends to clarify what exactly are the values in these figures (Legend of figures 2F and 4E):

2F “*Percentage of trigeminal neurons responding to both AITC and NTG vs total neurons responding to KCl, each data point is the percentage for one animal.*”

4E “*Right graph, 7% of neurons from trigeminal cultures of wild-type mice show transient currents after application of testosterone 10 μM and WS12 500 nM, while this response is abolished in TRPM8 knockout mice. Percentage of trigeminal neurons responding to both testosterone and WS12 vs total neurons responding to KCl, each data point is the percentage of one animal.*”

Titles have been added to these graphs in order to indicate that they are representations of the percentage of responsive neurons to the two stimuli applied:

2F

4E

How many neurons were measured in each condition?

We have included the number of cells analysed per animal in the Experimental Design subsection (Page 5, subsection “2.3. Experimental design”, included in the Supplementary Methods file). Each data point of the graphs showing the size of the calcium transients (2D, 3D) is the average normalized amplitude of the transients of responsive neurons from one animal. Each data point of the graphs showing percentage of neurons (2F, 4E) represents data from an individual animal. The number of neurons always ranged from 100 to 200 neurons per animal. Considering that the number of animals per condition was from 4 to 5, the total number of neurons measured per condition ranged from 400 to 1000. The number of neurons of calcium cellular experiments is now clarified in the Experimental Design subsection.

The legend states that the data points represent different experiments, does this mean different mice?

Yes, in those graphs each data point represents the value for one animal. These sentences in figures 2-4 (“Datapoints represent independent experiments”) were substituted by “*Data points without error bars represent values of individual animals*”, except for Figure 2I where the dots represent the quantified images (this was specified

in Figure 2 legend). The phrases included above were added to clarify the meaning of data points in the calcium imaging experiments (Legends of figures 2D, 2F, 3D and 4E):
 2D *“each data point represents the average response size of NTG-responsive neurons vs their respective KCl responses, in one animal.”*
 2F *“Percentage of trigeminal neurons responding to both AITC and NTG vs total neurons responding to KCl, each data point is the percentage for one animal.”*
 3D *“each data point represents the average response size of WS12-responsive neurons vs their respective KCl responses, in one animal.”*
 4E *“Percentage of trigeminal neurons responding to both testosterone and WS12 vs total neurons responding to KCl, each data point is the percentage of one animal.”*

It is stated that the responses to AITC are of longer duration in neurons of NTG-treated animals, but no statistics are given to back up this claim.

We removed “and duration” from the text (Results subsection 3.2, end of page 13) since there is not a formal quantification. The sentence is now focused on the amplitude of the response: *“These responses were of similar magnitude in samples of both sexes and had higher amplitude in neurons of mice chronically exposed to NTG”*

We also removed “and longer-lasting” from the discussion (page 19, lines 6-7 of 2nd paragraph of Discussion section) and now it reads:
“presented stronger TRPA1 activity in response to the specific agonist AITC”

I would like to indicate that none of these elements are crucially essential for the interpretations of the main results of this study, but it would be nice that they are considered to improve the quality of the manuscript.

Thanks for the suggestions, we included them and we believe quality is improved.

4.- Fig. 2D should show also the responses to high K+.

Responses to high K⁺ were added to Fig. 2C (Before Fig. 2D) to better represent the results:

5.- In Fig. 2F, the green traces are not clearly distinguishable from the black traces, please use a better color.

Green traces were substituted by fuchsia ones in Fig. 2E (Before Fig. 2F) and also to keep consistency in Fig. 4E. Bar plots on the sides also changed to fuchsia to keep consistency when labelling the cells of the KO animals.

Fig. 2E

Fig. 4E

6.- Lines 427-428, it is unclear what is meant by “...revealing that NTG sensitize through the activation of TRPA1”.

This phrase (now in page 14, 2nd paragraph) was modified to “*revealing an essential participation of TRPA1 in calcium transients elicited by both NTG and AITC in mice.*”

7.- Lines 467-468, it is stated that WS12 elicited Ca²⁺ transients with similar morphologies..., but no supporting data is provided. Visual inspection of the traces is not sufficient.

“*similar morphology and size*” (page 15, 1st paragraph) was substituted by “*similar amplitude*”

8.- Line 563, please delete “novel”. What is novel is the discovery, not the role of TRPM8. Please, also note that this sentence is too long.

This sentence (now in page 19, beginning of Discussion section) was split and modified as follows:

“The most salient contribution of our study is the discovery of **the** role of TRPM8 as an androgen-activated receptor that provides antinociceptive resilience **in a sex-dependent fashion. Specifically, TRPM8 activity** favours recovery **in a** mouse model of NTG-induced chronic migraine that produces similar acute mechanosensitivity in males and females but persistent **chronic** hypersensitivity exclusively in females.”

9.- Line 567, the statement “Testosterone by activating TRPM8...” and the

following sentence are too strong. I strongly suggest to write “Our results suggest that...”, or “We propose that...”.

We included these suggestions (page 19, 1st paragraph, lines 6-8):

“Our results suggest that testosterone by activating TRPM8 channels drives a sexual dimorphism characterized by recovery of normal sensitivity in males. We propose that the lack of this protective mechanism in females leads to a persistent mechanical hypersensitivity.”

10.- Line 576, a period is missing before “Overall...”.

Period added (page 19, right before last sentence of 1st paragraph of Discussion).

11.- Line 581, please delete the allusion to TRPA1 as thermoTRP. This may introduce an unnecessary element of confusion in the context of the present study.

“thermoTRP” changed to “*receptor*” (page 19, 1st sentence of 2nd paragraph of Discussion).

12.- Line 634, please write “AMTB”.

“*AMTB*” written (page 20, 1st line of last paragraph).

13.- Line 673, it is unclear what “TRPM8 testosterone activity” means. Testosterone-induced TRPM8 activation?

“*Testosterone-induced TRPM8 activation*” is now used (page 21, 3rd sentence of last paragraph).

Same for “calcium activity” in line 679. This was modified to “*lack significant effects on intracellular calcium imaging*” (page 21, lines 11-12 of last paragraph).

14.- The docking simulations should belong to the Results section.

Docking simulations were moved to the Results section, now subsection 3.6 of Results section (page 17, “**3.6 Computational docking studies in the menthol-binding site of TRPM8.**”)

15.- Lines 693-694, it is unclear to what “Additional less popular docking solutions” is referring to.

Our initial blind docking experiments on the homology models of rat and human TRPM8 consisted of 800 docking runs with flexible ligands (testosterone or WS12) that were clustered around binding hot spots as already described in Methods (Methods, page 10, subsection 2.14. Computational studies, 2nd paragraph). The best binding energy in each cluster was saved and the solutions grouped according to putative TRPM8 binding sites. These modelled solutions were arranged according to their frequency in the murine and human TRPM8 models. Only solutions with a frequency higher than 10% in at least one of the models were considered, obtaining docking solutions with lower frequency or popularity than the menthol binding site (Table 1, below).

Table 1. Putative binding sites found in murine and human TRPM8 models with the distributions of docking solutions.

1. Testosterone

Target regions	% Docking solutions (binding energy estimation, kcal/mol)	
	murine TRPM8	human TRPM8
Menthol binding site (S1-S2-S4-TRP domain)	39 (8.41)	20 (7.76)
Transmembrane, S1-S4-TRP domain	25 (7.58)	33 (7.84)
Pore, internal mouth	16 (7.65)	13 (8.15)
Transmembrane, S3-S4, S6	6 (7.94)	17 (8.15)

2. WS12

Target regions	% Docking solutions (binding energy estimation, kcal/mol)	
	murine TRPM8	human TRPM8
Menthol binding site (S1-S2-S4-TRP domain)	26 (7.20)	28 (7.51)
Transmembrane, S3-S4, S6	32 (7.24)	22 (6.93)
Pore, internal mouth	8 (6.56)	7 (6.75)
Transmembrane, S1-S4-TRP domain	8 (6.12)	11 (6.44)

While the alternative docking solutions aside of the menthol binding site may be of interest, there is a lack of knowledge about their impact on functionality, and mutagenesis studies would be required to characterize their actual relevance. Hence, to avoid being speculative, we decided to include only the results on the menthol binding site in the present study. Since we did not include these results, we are removing the phrase about the less popular solutions from the discussion section to increase clarity (page 22, end of the paragraph before the last paragraph of the discussion): ~~Additional less popular docking solutions include regions without described impact on TRPM8 functionality, and mutagenesis studies will be needed to clarify the specific role of these interactions.~~

16.- How dose activation of TRPM8 result in reduced mechanical hypersensitivity?

The inhibition of mechanical hypersensitivity through TRPM8 activation may involve the recruitment of inhibitory signalling at the spinal cord level and interactions with the endogenous opioid system. This has been discussed as follows in Discussion section, end of page 19- beginning of page 20 (already indicated above):

“Several mechanisms could involve TRPM8-mediated antinociception. On the one hand, it has been proposed that TRPM8 activation produces silencing of C nociceptors expressing TRPA1 (Liu et al., J Appl Physiol 2015) through interneuron recruitment in the spinal cord dorsal horn (Dussor and Cao, Headache 2016; Kayama et al., Cephalalgia 2018; Ren, Dhaka, and Cao, Mol. Pain 2015) or via inhibitory metabotropic glutamatergic receptors expressed in the central terminal of nociceptors (Proudfoot et al., Curr Biol 2006). Thus, in the studied pain models TRPM8 activity may occlude signalling of TRPA1-expressing nociceptors and subsequent CGRP release (Liu et al., J Appl Physiol 2015). In addition, a participation of central brain areas cannot be excluded

from our data, given the recent description of TRPM8 also in supraspinal locations (Ordás, *J Comp Neurol* 2021).”

And in page 21, 1st paragraph:

“In this context, an interdependency between TRPM8 and the endogenous opioid system has been described, where TRPM8 knockout mice display reduced morphine analgesia and the antinociceptive effects of TRPM8 agonists being sensitive to the opioid antagonist naloxone (Shapovalov et al., 2013; Liu et al., 2013). Our results suggest the interest of investigating opioid-TRPM8 interactions for the promotion of pain resilience.”

17.- Can the authors make any “biological sense” of the sexual dimorphism of chronic migraine? Are females just doomed to suffer from chronic pain by their lack of testosterone?

Increased pain sensitivity of murine and primate females could reduce the likelihood of aggressive behaviours and prevent further injuries inside the social structures of these species, which normally involve a relatively harmonious coexistence of multiple females. In addition, it could engage protective and sedentary behaviours directed to favour success in reproductive processes such as fertilization or embryo implantation. In contrast, murine and primate males are highly territorial and develop strong hierarchical structures that involve frequent exposure to fights and injuries. The existence of a protective mechanism such as testosterone-induced TRPM8 activation could promote fast recovery of normosensitivity in males in order to be ready for the following threat.

Fortunately, the understanding of the physiological and molecular causes of the sex differences in pain perception will allow the development of novel, specific and safer chronic pain treatments for women and men.

If considered appropriate, the paragraph on the “biological sense” could be added to the discussion (Last paragraph, page 22).

18.- Do the authors expect that their results can be extended to other TRPA1-mediated chronic pains?

In our experimental conditions, both chronic NTG and acute formalin pain models were antagonized by TRPM8 activation. Since formalin also requires the presence of TRPA1 (McNamara et al., 2007) and is similarly able of inducing chronic pain sensitization characterized by enhanced mechanosensitivity (Martínez-Rojas et al., 2017), we do expect that the results can be extended to other TRPA1-mediated chronic pains.

- ✓ Martínez-Rojas VA, García G, Noriega-Navarro R, et al. Peripheral and spinal TRPA1 channels contribute to formalin-induced long-lasting mechanical hypersensitivity. *J Pain Res.* 2017;11:51-60. Published 2017 Dec 27. doi:10.2147/JPR.S153671
- ✓ McNamara CR, Mandel-Brehm J, Bautista DM, et al. TRPA1 mediates formalin-induced pain. *Proc Natl Acad Sci U S A.* 2007;104(33):13525-13530. doi:10.1073/pnas.0705924104.

Reviewer #3 (Remarks to the Author):

It would be nice to see further validation of this murine migraine model. For instance, instead of measuring solely mechanical sensitivity, one could also measure cortical spreading depression.

Unfortunately, we don't have the infrastructure needed to perform optical or electrophysiological in-vivo measurements of the parameters associated with cortical spreading depression. However, we gathered additional validation data of the sexual dimorphism of this murine migraine model based on the relative weight of the adrenal glands of males and females exposed to chronic nitroglycerin (NTG). In this context, the stress response has been involved into the pathophysiology of chronic migraine (Stubberud et al., 2021) and glucocorticoid/mineralocorticoid responses could be relevant phenotypes that merit further investigation.

In our experimental conditions, wild-type female mice subjected to the model of chronic migraine and sacrificed 14 days later develop a significant increase of the adrenal weight /body weight ratio when they are exposed to nitroglycerin (Graph A below, Female Left (L) and Right (R) adrenal glands, where black star means $p < 0.05$ NTG vs. Vehicle, $n = 6-7$ per group). On the other hand, male mice are heavier and have smaller glands than females (Graphs B, C) and their adrenal weight/body weight ratio is similar regardless of the treatment received (Graph A, Nonsignificant differences NTG vs. Vehicle, $n = 6-7$ per group). Nonsignificant changes associated with the treatments are found in body weights or raw adrenal weights (Graphs B and C below). While adrenal glands are closely related to the pathophysiology of chronic migraine, these findings seem out of the scope of the present investigation and we would like to preserve the data for future work.

- ✓ Stubberud A, Buse DC, Kristoffersen ES, Linde M, Tronvik E. Is there a causal relationship between stress and migraine? Current evidence and implications for management. *J Headache Pain*. 2021;22(1):155. Published 2021 Dec 20. doi:10.1186/s10194-021-01369-6

The claim put forth that androgenic TRPM8 activity drives sexual dimorphism of chronic migraine has not been substantiated. Given the controversy surrounding whether testosterone is a TRPM8 agonist, additional experiments are required to demonstrate that testosterone binds directly to TRPM8 (and that the effects are not indirect). Knock-out mice and AMTB inhibition show that TRPM8 is linked to the behavioural phenotype, but not that TRPM8 is a testosterone receptor. While calcium imaging experiments have been conducted (with modest effect), electrophysiology would be far more convincing.

We conducted electrophysiological experiments to study the response to testosterone in HEK293 cells heterologously expressing human TRPM8 (hTRPM8). Current density (pA/pF) vs voltage relationships in these cells (Left graph below) evidenced an increase in outward rectifying current after application of 10 pM testosterone. This testosterone-induced current ($P < 0.001$ vs. Basal current) was significantly blocked in the presence of 10 μ M AMTB (Left graph and right graph showing current at +80 mV, $P < 0.001$ vs. Testosterone + AMTB vehicle). These results were added to the results section (page 17, 3.5. subsection) and are part of the new Figure 5 (Figure 5A-B). Methods were also included (page 10, subsection 2.16. Patch-clamp whole cell recordings)

Computational docking studies based on homology models of a 4.1 Å structure are not suitable to demonstrate ligand binding, especially given that this pocket has been shown to be basically nondiscriminatory. It should be noted that both progesterone and estradiol were also able to be positioned in the cavernous and adaptable ligand-binding pocket of TRPM8.

This was noted in the results section (page 18, 3.6. subsection, end of last paragraph):

“Additional local experiments analysed TRPM8 affinity for the female hormones progesterone and estradiol and both hormones were also able to be positioned in the cavernous and adaptable ligand-binding pocket of TRPM8, revealing binding energies of 8.16 and 8.97 kcal/mol in the rat (progesterone and estradiol respectively, Figures 6C-D) and of 9.37 and 8.15 in the human TRPM8 models (Supplementary Figure 5C-D).”

As testosterone’s primary mechanism of action is through binding to the androgen receptor, should androgen receptor knock-out mice not also be evaluated?

Androgen receptor (AR) knockout mice (Yeh et al., 2002) show developmental alterations such as female-like phenotype in males and reproductive deficits in females, while possible alterations on neuronal TRPM8 expression have not been explored. To circumvent these issues, and to study the impact of AR expression on TRPM8 function, we assessed testosterone-induced currents in hTRPM8 HEK cells transfected with small interfering RNA (siRNA) against AR (ThermoFisher, Ambion, s1538) and in control

hTRPM8 HEK cells transfected with scrambled siRNA (Thermo Fisher, Ambion, 4390843). Before conducting the electrophysiological recordings, hTRPM8 HEK cells were transfected for 72h and western blot for the AR was conducted to confirm the knockdown of the AR. As shown in the picture below, anti-AR siRNA efficiently disrupted AR expression whereas hTRPM8 HEK cells showed evident expression. Equal protein loading was ensured through assessment of β -tubulin expression (below).

Once the efficiency of the AR knockdown was confirmed, new transfections were conducted in hTRPM8 HEK cells to assess the electrophysiological response to testosterone. Wild-type HEK cells (WT HEK) without heterologous hTRPM8 expression were cultured in parallel. In these experimental conditions, hTRPM8 HEK cells treated with either anti-AR siRNA or scrambled siRNA showed similar current density-voltage curves after 10 pM testosterone, whereas WT HEK cells did not have testosterone-induced currents as shown in the graphs below ($P < 0.01$ WT HEK vs hTRPM8 HEK, current fold increase at +80 mV).

Western blot of AR and β -tubulin loading control in cells cultured in parallel to the ones used for the electrophysiological studies showed again disrupted AR expression in hTRPM8 HEK transfected with anti-AR siRNA, when compared to scrambled siRNA hTRPM8 HEK cells or WT HEK cells. Hence, down-regulation of the AR did not affect testosterone-induced currents in HEK 293 cells expressing human TRPM8.

These results were included in the manuscript as Figures 5C-E, and were added to the results section (page 17, Results subsection 3.5. Testosterone-induced current of human TRPM8 is independent of androgen receptor expression). Procedures for

transfections and western blot were also incorporated to the Methods section (page 10, subsection **2.17. Knockdown of androgen receptor in HEK 293 cells expressing human TRPM8**.; pages 10-11, subsection **2.18. Protein extraction and western blotting**).

✓ Yeh S, Tsai MY, Xu Q, et al. Generation and characterization of androgen receptor knockout (ARKO) mice: an in vivo model for the study of androgen functions in selective tissues [published correction appears in Proc Natl Acad Sci U S A 2002 Nov 12;99(23):15245. Hung Min-Chi [corrected to Hung Mien-Chie]]. *Proc Natl Acad Sci U S A*. 2002;99(21):13498-13503. doi:10.1073/pnas.212474399

The scientific foundation of this study (that TRPM8 is a testosterone receptor) is not solid enough to allow for the numerous assumptions that have been made in interpreting the data. Assembling this data into a manuscript with a different focal point is suggested.

Assumptions considering TRPM8 as a testosterone receptor were softened:

Abstract, page 2, last sentence:

“highlights the interest of molecular solutions mimicking the pain-relieving activity of testosterone on TRPM8” changed to *“highlight the interest of molecular solutions mimicking the pain-relieving activity of testosterone through TRPM8”*

Discussion, page 19, last lines, *“high-affinity”* was removed from *“testosterone exerts an antinociceptive role through high-affinity interactions with TRPM8”*.

Discussion, pages 21-22: *“the results obtained both in murine and human TRPM8 models indicate that testosterone most likely binds to the active pocket described for WS12 and menthol”* were changed to *“the results obtained both in murine and human TRPM8 models suggest binding of testosterone to the active pocket described for WS12 and menthol”*

In Discussion, page 22, end of 1st paragraph: *“Our data suggest a direct testosterone-TRPM8 interaction”* was changed to *“Our data indicate a testosterone-TRPM8 interaction”*

REVIEWER COMMENTS

Reviewer #2 (Remarks to the Author):

The authors have provided appropriate responses to my previous comments. I have only two minor points:

- 1.- The legend of figure 4E refers to "currents", but this is a Ca²⁺ imaging experiment in which no currents are actually measured.
- 2.- Please, include a discussion about the a possible biological significance of the findings (point 17 of my previous evaluation).

Reviewer #3 (Remarks to the Author):

I appreciate the considerable thought and work that has gone into this manuscript however my primary concern remains that the scientific foundation of this study (that TRPM8 is a testosterone receptor) is not solid enough to allow for the numerous assumptions that have been made in generating hypotheses and interpreting the data. I appreciate that the authors' have introduced whole cell patch-clamp electrophysiology data but find it difficult to interpret with no comparison to an established TRPM8 antagonist, such as menthol, WS-12, or cold. The fact that computational docking studies based on homology models of a 4.1 Å structure are not suitable to demonstrate ligand binding, especially given that this pocket has been shown to be basically nondiscriminatory, has not been adequately addressed. Assembling this data into a manuscript with a different focal point is still what I suggest.

REVIEWER COMMENTS

Reviewer #2 (Remarks to the Author):

The authors have provided appropriate responses to my previous comments. I have only two minor points:

1.- The legend of figure 4E refers to "currents", but this is a Ca²⁺ imaging experiment in which no currents are actually measured.

"calcium currents" was changed to "calcium transients" in figure legends 4E and 3D

2.- Please, include a discussion about the a possible biological significance of the findings (point 17 of my previous evaluation).

The discussion about possible biological significance (point 17 of previous evaluation) was included in the last paragraph of Discussion section (page 22, lines 25-35):

"Increased pain sensitivity of murine and primate females could reduce the likelihood of aggressive behaviours and prevent further injuries inside the social structures of these species, which normally involve a relatively harmonious coexistence of multiple females. In addition, it could engage protective and sedentary behaviours directed to favour success in reproductive processes such as fertilization or embryo implantation. In contrast, murine and primate males are highly territorial and develop strong hierarchical structures that involve frequent exposure to fights and injuries. The existence of a protective mechanism such as testosterone-induced TRPM8 activation could promote fast recovery of normosensitivity in males in order to be ready for the following threat."

Reviewer #3 (Remarks to the Author):

I appreciate the considerable thought and work that has gone into this manuscript however my primary concern remains that the scientific foundation of this study (that TRPM8 is a testosterone receptor) is not solid enough to allow for the numerous assumptions that have been made in generating hypotheses and interpreting the data. I appreciate that the authors' have introduced whole cell patch-clamp electrophysiology data but find it difficult to interpret with no comparison to an established TRPM8 antagonist, such as menthol, WS-12, or cold.

As suggested, we have conducted the additional electrophysiological experiments to strengthen the part of the study describing the potential interaction of testosterone with TRPM8. We conducted dose-response curves with testosterone (EC₅₀ 7 pM) and the potent and selective menthol derivative WS12 (EC₅₀ 500 nM) in the HEK cells heterologously expressing TRPM8 (New Supplementary Figure 5A, below). Afterwards, we used 500 nM WS12 (EC₅₀) to determine the IC₅₀ of the TRPM8 antagonist AMTB (IC₅₀=15 nM) which binds also to the menthol-binding site (New Supplementary Figure 5B). Application of AMTB 15nM shifted the dose-reponse curve of testosterone to the right (New Figure 5F), being compatible with competitive inhibition.

Supplementary Figure 5A

Supplementary Figure 5B

Figure 5F

These results were included at the end of Results section 3.5 (page 17, lines 26-37):

“In order to allow the comparison to an established TRPM8 agonist in the same setting, we conducted dose-response curves for testosterone and WS12 in untreated hTRPM8-HEK293 cells, obtaining EC₅₀s of 7 pM for testosterone and 500 nM for WS12 (Supplementary Figure 5A). The EC₅₀ of WS12 was afterwards used to estimate the IC₅₀ of the TRPM8 antagonist AMTB, which also binds to the menthol-binding site (Diver, Cheng and Julius, Science 2019) (Supplementary Figure 5B). Subsequent application of AMTB IC₅₀ (15 nM) shifted the testosterone dose-response curve to the right (Figure 5F, EC₅₀=34 pM), being compatible with competitive inhibition. Hence, testosterone-induced currents were selectively elicited in HEK293 cells heterologously expressing human TRPM8, were independent of AR expression and sensitive to the TRPM8 antagonist AMTB, substantiating an ionotropic action of the hormone.”

And a sentence was modified in the discussion (page 12, lines 40-43):

“Whole cell patch-clamp recordings in HEK cells heterologously expressing human TRPM8 also revealed testosterone-induced currents independently of androgen receptor expression and sensitive to the TRPM8 antagonist AMTB.”

The fact that computational docking studies based on homology models of a 4.1 Å structure are not suitable to demonstrate ligand binding, especially given that this pocket has been shown to be basically nondiscriminatory, has not been adequately addressed. Assembling this data into a manuscript with a different focal point is still what I suggest.

We concur with the referee that computational docking is a supportive methodology to guide experimental design. We used to interrogate if our experimental results are consistent with a direct interaction of testosterone with the receptor, in agreement with results from other groups (Asuthkar et al. (2015) J. Biol. Chem. 290, 2670-2688 ; D'Arrigo et al. (2021) Molecules. 26(6):1613). Taken together, our findings substantiate the proposed tenet that testosterone ionotropically activate TRPM8, consistent with the presence of binding site, most likely located near the menthol binding site.

We have carefully revised the manuscript to emphasize that our focal point is to convey the message that the sex dimorphism observed in the migraine model appears mediated by testosterone signaling through the TRPM8 receptor, possibly through an ionotropic mechanism, although demonstration of a direct binding of the androgen to the channel requires additional experiments. Indeed, either direct binding measurements in highly purified receptors or solving the atomic structure of TRPM8 bound to testosterone will be needed to confirm a direct binding and to unveil the receptor binding site.

The following sentence was included in the results section (page 18, lines 18-25) to address the referee concern. *“Thus, local docking of testosterone in the menthol binding pocket suggests an interaction that is consistent with the potent EC50 obtained (Supplementary Figure 5A), further suggesting a potential ionotropic mechanism of the hormone on TRPM8 channels. However, it should be noted that molecular modeling using homology models at 4.1 Å resolution are limited to unequivocally demonstrate ligand binding and experimental studies are required to assess a direct binding of testosterone to TRPM8 and to identify the binding pocket.”*

And in the discussion section (page 22, lines 4-5), we also included the sentence *“TRPM8 models suggest binding of testosterone to the active pocket described for WS12 and menthol^{55,81}, that will need additional experiments for confirmation”*

REVIEWER COMMENTS

Reviewer #3 (Remarks to the Author):

The patch-clamp electrophysiology data demonstrating that testosterone activates TRPM8 has been greatly expanded upon. However, it is quite complexing. The testosterone EC50 of 7 pM seems much too low to be consistent with what we know about TRPM8 physiology. For instance, the EC50 of menthol is in the low micromolar range. The author's results for WS12 are an EC50 of 500 nM. Even the super-cooling agent, icilin, only has an EC50 of 360 nM. Also, why are the inhibitory effects of AMTB so limited? What is its IC50 with respect to testosterone? A discussion surrounding these ideas should be included.

I remain unconvinced by the author's rebuttal that the docking studies support their conclusions. If testosterone is a ligand for TRPM8, then the female hormones, progesterone and estradiol, should serve as negative controls, as they have a similar structure but no effect on the ion channel. However, they both dock to the identical pocket in TRPM8 with similar binding coefficients. (Or perhaps, I have misunderstood, and the authors are proposing that numerous hormones are ligands for TRPM8. If so, for female hormones, is this not inconsistent with your narrative that men have a protective migraine pain advantage? Also, do you have any indication, published or unpublished, that progesterone and estradiol are ligands for TRPM8?) This likely stems from the fact that any hydrophobic molecule of a particular size would dock into these low-resolution human and mouse homology models. Figure 6 and the associated text should be removed from this manuscript.

REVIEWER COMMENTS

Reviewer #3 (Remarks to the Author):

The patch-clamp electrophysiology data demonstrating that testosterone activates TRPM8 has been greatly expanded upon. However, it is quite complexing. The testosterone EC₅₀ of 7 pM seems much too low to be consistent with what we know about TRPM8 physiology. For instance, the EC₅₀ of menthol is in the low micromolar range. The author's results for WS12 are an EC₅₀ of 500 nM. Even the super-cooling agent, icilin, only has an EC₅₀ of 360 nM.

We concur with the reviewer that it is intriguing the relatively low EC₅₀ for testosterone when compared to that of exogenous compounds such as menthol, WS12 or icilin. However, these concentrations are in the range of those reported to evoke ionic currents in TRPM8 channels incorporated into planar lipid bilayers (Mohandass et al., FASEB J 2020 doi:10.1096/fj.202000794R Fig S4) and are similar to those previously described by Asuthkar et al. in prostate cancer cells and neuronal cells (J Biol Chem. 2015 doi:10.1074/jbc.M114.610873). In this line, picomolar concentrations of other endogenous molecule such as oxytocin have shown agonistic effects over TRPV1 receptors (Nersesyan et al., Cell Rep 2017 doi:10.1016/j.celrep.2017.10.063). On the other hand, these concentrations are different to the nanomolar concentrations provoking TRPM8 inhibition that have been described for HEK293 and neuronal cells (Gkika et al., FASEB J 2020, doi:10.1096/fj.201902270R). Given that testosterone-induced currents can be substantially reduced by plasma proteins (Mohandass et al., FASEB J 2020 doi:10.1096/fj.202000794R Fig S4), it could be possible that low picomolar testosterone concentrations are sufficient to trigger TRPM8-related activity in neurons and also in other cell types such as prostate or liver cells, where TRPM8 shows much higher expression when compared to neural tissues (Uhlén et al., Science 2015 doi:10.1126/science.1260419, www.proteinatlas.org). Interestingly, picomolar concentrations of 5 α -dihydrotestosterone (DHT) induced TRPM8 overexpression in cancer cells, whereas such overexpression was much lower also after exposure to higher DHT concentrations (Bidaux et al., Endocr Relat Cancer. 2005, doi:10.1677/erc.1.00969), suggesting inverted u-shaped dose-effect curves for these types of steroids. Notably, this type of physiological behavior has been also observed in steroid-like molecules acting as endocrine disruptors such as bisphenol A, that exhibits significant activity at very low concentrations but not at higher concentrations (Alonso-Magdalena et al., Mol Cell Endocrinol 2012, doi:10.1016/j.mce.2011.12.012).

Also, why are the inhibitory effects of AMTB so limited? What is its IC₅₀ with respect to testosterone?

As the referee comments, the inhibitory potency of AMTB is apparently limited attending to the effect on the testosterone dose-response. However, it should be noted that for evaluating a competitive binding mechanism it is normally used a concentration of antagonist (AMTB) around its IC₅₀, as this represents a concentration that blocks half of the agonist-elicited response. Higher concentrations of the antagonists lead to higher blockade and lower ionic currents that may compromise the accuracy of current measurements. In our case, even using the IC₅₀ for AMTB (15 nM) determined with respect to WS12 (that is similar to other channel agonists such as icilin and menthol), we observed a notable 5-fold decrease in testosterone potency activating TRPM8 (EC₅₀ change from 7 to 34 pM), which represents a significant right-shift of the dose-response curve. It is expected that higher AMTB concentrations within the nanomolar-micromolar range should prevent TRPM8 activation by picomolar testosterone as suggested now in the discussion section (Page 20, lines 48-49). Indeed, a 10 μ M AMTB fully blocked testosterone agonistic activity (Figs. 5A and B).

A discussion surrounding these ideas should be included.

The following text was added/modified in the discussion section (last paragraph of page 20, first one of page 21):

“Our trigeminal cultures revealed TRPM8 calcium transients in response to picomolar concentrations of testosterone⁸⁰ that were absent in cultures of TRPM8 knockouts. Whole cell patch-clamp recordings in HEK cells heterologously expressing human TRPM8 also revealed ionic currents at low picomolar testosterone concentrations (EC50=7pM), in contrast to the lower potency of WS12 (EC50=500 nM) obtained in the same experimental conditions. These testosterone-induced currents were independent of androgen receptor expression and sensitive to the TRPM8 antagonist AMTB, as revealed by the 5-fold decrease in testosterone potency (EC50 increased from 7 to 34 pM) in the presence of 15 nM AMTB (IC50 of AMTB against the less potent WS12). Note that the high potency of testosterone evoking TRPM8 ionic currents, requires higher AMTB concentrations to abolish testosterone-induced TRPM8 responses. Indeed, 10 μM AMTB fully blocked testosterone-evoked ionic currents. In agreement with our data, other groups have revealed TRPM8-mediated currents after application of picomolar concentrations of testosterone to cancer cells, somatosensory neurons²⁸ or planar lipid bilayers expressing TRPM8⁸¹. Thus, testosterone induced rapid non-genomic effects on primary afferent neurons and other cell types through TRPM8. Similarly, other endogenous molecules such as oxytocin have shown agonistic effects over TRPV1 also at picomolar concentrations⁸². While WS12 in the nanomolar range elicits stronger calcium transients than picomolar concentrations of testosterone, at the picomolar level TRPM8 agonists such as menthol, icilin or WS12 lack significant effects on intracellular calcium imaging²⁸. On the other hand, nanomolar concentrations of testosterone inhibited TRPM8⁸³ in HEK cells and neurons, suggesting possible inverted u-shaped dose-effect curves for this type of steroids. In agreement, picomolar concentrations of dihydrotestosterone induced pronounced overexpression of TRPM8 in cancer cells whereas higher concentrations had a reduced effect⁸⁴.”

These references were added:

- Bidaux G, Roudbaraki M, Merle C, et al. Evidence for specific TRPM8 expression in human prostate secretory epithelial cells: functional androgen receptor requirement. *Endocr Relat Cancer*. 2005;12(2):367-382. doi:10.1677/erc.1.00969
- Gkika D, Lolignier S, Grolez GP, et al. Testosterone-androgen receptor: The steroid link inhibiting TRPM8-mediated cold sensitivity. *FASEB J*. 2020;34(6):7483-7499. doi:10.1096/fj.201902270R
- Mohandass A, Krishnan V, Gribkova ED, et al. TRPM8 as the rapid testosterone signaling receptor: Implications in the regulation of dimorphic sexual and social behaviors. *FASEB J*. 2020;34(8):10887-10906. doi:10.1096/fj.202000794R
- Nersesyan Y, Demirkhanyan L, Cabezas-Bratesco D, et al. Oxytocin Modulates Nociception as an Agonist of Pain-Sensing TRPV1. *Cell Rep*. 2017;21(6):1681-1691. doi:10.1016/j.celrep.2017.10.063

I remain unconvinced by the author’s rebuttal that the docking studies support their conclusions. If testosterone is a ligand for TRPM8, then the female hormones, progesterone and estradiol, should serve as negative controls, as they have a similar structure but no effect on the ion channel. However, they both dock to the identical pocket in TRPM8 with similar binding coefficients. (Or perhaps, I have misunderstood, and the authors are proposing that numerous hormones are ligands for TRPM8. If so, for female hormones, is this not inconsistent with your narrative that men have a

protective migraine pain advantage? Also, do you have any indication, published or unpublished, that progesterone and estradiol are ligands for TRPM8?) This likely stems from the fact that any hydrophobic molecule of a particular size would dock into these low-resolution human and mouse homology models.

We believe that the docking of estradiol or progesterone in the same pocket is not necessarily inconsistent with the protective phenotype of males associated with testosterone. In women, there is migraine pain associated with the menstruation, which coincides with a drop in estrogen levels. Also, a protective effect of progesterone has been widely described in different pain models, which agrees with the low incidence of migraine during pregnancy. Thus, possible interactions of TRPM8 with these molecules need to be studied in future investigations. A published paper (Mohandass et al., FASEB J 2020 doi:10.1096/fj.202000794R Fig 3C) indicates progesterone and estradiol-induced currents in planar lipid bilayers exclusively expressing TRPM8 (EC50s of 0.5 and 1.2 μ M), although with lower potency than testosterone (E50=22.4 pM) or dihydrotestosterone (EC50=23.5 nM). In this case, dihydrotestosterone, more hydrophobic than testosterone, shows much lower potency than testosterone.

Figure 6 and the associated text should be removed from this manuscript.

We have considered the referee concern of using low resolution models of TRPM8 for mapping drug binding sites in the channel structure and using those for supporting our experimental results. Thus, following the referee suggestion we have removed Figure 6 and the associated text from the results section. However, we believe that docking experiments in receptor homology models, for which we do not have yet structural information, provide testable models that can further our knowledge of ligand-receptor interactions. In this regard, we included a comment in the discussion section indicating that these models could be used as testable hypothesis in future investigations directed to identify molecular determinants of the putative testosterone binding site. Thus, for illustration, we included a computational homology model of the human TRPM8 docked with WS12 and the sexual steroids in the supplementary information (Supplementary Figure 6). We trust the referee will consider the use of this hypothetical model as appropriate for discussion purposes.

The following text is included in the discussion section (page 21, lines 16-31):

“Our calcium imaging, electrophysiological and behavioral results suggest an ionotropic effect of testosterone on TRPM8 channels and hint to the presence of a potential hormone binding site within the receptor structure. Docking simulations on a computational homology model of the human TRPM8 suggest the presence of a putative testosterone binding site in the receptor transmembrane domain with a theoretical binding energy and affinity consistent with the hormone potency (Supplementary Figure 6). Furthermore, this putative ligand-receptor complex model suggests binding of testosterone to the active pocket described for WS12 and menthol^{70,85}. Interestingly, female hormones such as estradiol or progesterone were also able to be positioned in this binding pocket (Supplementary Figure 6) although with lower binding energy estimates than testosterone. This agrees with the potency of estradiol and progesterone evoking TRPM8 ionic currents in planar lipid bilayers⁸¹, as compared with testosterone. This computational docking model provides a testable hypothesis for future studies aimed to identify the molecular determinants of this putative binding site, either by structural and/or by structure-function studies.”

Methods for the computational docking simulations using the model of human TRPM8 have been moved to supplementary methods.

REVIEWERS' COMMENTS

Reviewer #3 (Remarks to the Author):

The authors have done a good job of addressing the critiques.

REVIEWERS' COMMENTS

Reviewer #3 (Remarks to the Author):

The authors have done a good job of addressing the critiques.

Thanks for your feedback.